# Annual time-series analysis of aqueous eDNA reveals ecologically relevant dynamics of lake ecosystem biodiversity

Iliana Bista[1], Gary R. Carvalho[1], Kerry Walsh[2], Mathew Seymour[1], Mehrdad Hajibabaei[3], Delphine Lallias[4], Martin Christmas[2] & Simon Creer[1]

The use of environmental DNA (eDNA) in biodiversity assessments offers a step-change in sensitivity, throughput and simultaneous measures of ecosystem diversity and function. There remains, however, a need to examine eDNA persistence in the wild through simultaneous temporal measures of eDNA and biota. Here, we use metabarcoding of two markers of different lengths, derived from an annual time series of aqueous lake eDNA to examine temporal shifts in ecosystem biodiversity and in an ecologically important group of macro-invertebrates (Diptera: Chironomidae). The analyses allow different levels of detection and validation of taxon richness and community composition (β-diversity) through time, with shorter eDNA fragments dominating the eDNA community. Comparisons between eDNA, community DNA, taxonomy and UK species abundance data further show significant relationships between diversity estimates derived across the disparate methodologies. Our results reveal the temporal dynamics of eDNA and validate the utility of eDNA metabarcoding for tracking seasonal diversity at the ecosystem scale.

[1] Molecular Ecology and Fisheries Genetics Laboratory, School of Biological Sciences, Bangor University, Bangor, Gwynedd LL57 2UW, UK. [2] Environment Agency, Horizon House, Deanery Road, Bristol BS1 5AH, UK. [3] Department of Integrative Biology & Biodiversity Institute of Ontario, University of Guelph, Guelph, Ontario, Canada N1G 2W1. [4] GABI, INRA, AgroParisTech, Université Paris-Saclay, 78350 Jouy-en-Josas, France. Correspondence and requests for materials should be addressed to S.C. (email: s.creer@bangor.ac.uk) or to I.B. (email: ilianabista@gmail.com).

The maintenance of biodiversity underpins the stability of ecosystem processes in constantly changing environments[1]. Consequently, biodiversity loss not only affects ecosystem function and services, but also society as a whole[2]. One major impediment for elucidating the relationship between biodiversity and ecosystem health is a need for robust and detailed understanding of biodiversity processes and dynamics in time and space[3]. To halt or reverse contemporary species loss and habitat degradation, there is a need for increasingly reliable and cost effective methods for biodiversity assessment, since widely employed traditional approaches fall short in many cases[4]. Currently, species identification of individuals at immature life stages and among closely related species is difficult and requires high-level, labour-intensive taxonomic expertise, thereby rendering large scale ecosystem-wide assessments expensive, time consuming and potentially unrepresentative of the ecosystem sampled[5]. However, recent advances in molecular detection techniques, most notably the application of environmental DNA (eDNA), offer exciting new opportunities to improve existing biodiversity assessment procedures.

Environmental DNA (eDNA) is DNA extracted directly from an environment sample (for example, water, soil or air), without prior isolation of the organisms themselves[6]. Sources of eDNA include sloughed skin cells, urine, faeces, saliva or other bodily secretions[7], and consist of both free molecules (extracellular DNA) and free cells[8]. Furthermore, eDNA collected from water samples has highly sensitive detection capability and is non-invasive to the sampled biota[9], thereby potentially improving environmental management and assessment of freshwater ecosystems[4,10].

Previous work with eDNA of aquatic invertebrates is dominated by PCR-based approaches, which are limited in assessing biodiversity[11–13]. However, high throughput sequencing (HTS) applications, such as metabarcoding, are already advancing prospects in ecology[14], offering comprehensive and efficient tools for measuring and assessing total biodiversity[15]. High throughput sequencing has successfully been used for sequencing whole communities of invertebrates (bulk samples)[16–18], though only a few studies have employed metabarcoding of aqueous eDNA[19,20], and even fewer for invertebrates[3]. Most aqueous eDNA studies have focused on macro-organisms, including fish and amphibians[19–21], with limited focus on arthropods[22,23]. Nevertheless, the combination of HTS and eDNA is poised to become a prominent tool for ecosystem assessment[10,22] by simultaneously assessing a plethora of organisms, including associated organism interactions, with a throughput sufficient for rapid whole community assessment.

Regardless of the increasing number of eDNA studies, several factors of eDNA research demand clarification, including persistence of eDNA[24]. Persistence of eDNA is the time that eDNA remains detectable (for example, in the water) after removal or loss of the organism from the environment, which influences the timeframe for biodiversity assessment[6]. Investigating the temporal relationship between community DNA[25] and eDNA is vital, since accurate (extant) biodiversity assessment requires detection of contemporary, and ecologically relevant, biodiversity. The persistence of eDNA for several different species has been studied mainly in artificial systems, including aquaria and mesocosms[6,11,22,26]. Notably, persistence of short eDNA fragments, in artificial environments, was found to vary between days to weeks after removal of the study organisms, depending upon biotic and abiotic factors[27].

Species identity by eDNA is typically undertaken by detection of short DNA fragments[7], a practise possibly influenced by ancient DNA work, which utilizes highly fragmented DNA[28]. For the detection of rare and evasive species, short DNA fragments might indeed increase detection, although with some risk of errors if not properly analysed. Possible biases when using short fragments include inadvertently sampling old eDNA fragments which have demonstrated remarkable persistence[8], especially when bound to sediments where the degradation rate is slower, due to protection of DNA molecules and inactivation of extracellular nucleases[27]. Conversely, DNA fragments of several hundred base pairs length are less likely to persist after release into the environment due to rapid degradation[29] and may represent a less abundant, but more contemporary, biodiversity signal[30].

While the ecological value of collecting temporal data is established, most ecological studies focus on spatial data[31]. Similarly, many existing eDNA studies have focused on spatial detection, such as early detection of invasive species[11,32] and presence of rare, or endangered species[33]. Temporal estimates have been relatively neglected in eDNA studies (but see ref. 33 for repeated seasonal sampling), and an understanding of temporal relationships between eDNA and community biodiversity remains a key knowledge gap[3]. In addition there are no published studies, to our knowledge, employing temporally collected data that incorporate seasonal variation across an annual cycle from aqueous eDNA for ecosystem-wide biodiversity level analysis.

Furthermore, overall ecosystem biodiversity characterization, using indicator taxonomic groups, can facilitate comparisons between taxonomically identified biodiversity over time (for example, collection of invertebrate samples) and eDNA detection. One such indicator group is the Chironomidae or non-biting midges (Diptera: Chironomidae), which exhibit specialized responses to ecological stressors and are acknowledged as one of the most informative macroinvertebrate groups for monitoring lake ecosystem health[34,35]. Importantly, samples can be collected after adult emergence in the form of shed skins of the pupae (pupal exuviae) that float on the water surface. The exuviae technique allows for integrated sampling of lake ecosystems from all aquatic microhabitats of the lake, and sample identification can yield insights on ecosystem-wide biodiversity[34].

Accordingly, here we (a) investigate whether metabarcoding of lake eDNA is effective for the detection of community diversity and temporal shifts in an ecologically important sentinel group of macroinvertebrates, via comparison with the molecular and morphological analysis of chironomid exuvial bulk samples; (b) investigate the use of eDNA analyses for characterizing whole-ecosystem biodiversity patterns; and (c) explore the effects of amplicon length on detection of contemporary diversity. We show that freshwater lake eDNA analyses capture seasonally coherent biodiversity patterns across the tree of life and that shorter fragments of eDNA dominate natural ecosystems. Moreover, species incidence measured by metabarcoding of eDNA and DNA derived from communities overlap substantially with traditional taxonomic assessment. Collectively, we examine the ecological relevance of eDNA by exploring mechanisms underpinning the temporal dynamics of eDNA and the biological community at the ecosystem scale in nature.

## Results

**Sequencing results**. After stringent filtering and quality control, 13,100,236 reads were obtained for: (1) the full-length COI barcoding region (658 bp) (amplicon COIF 6,659,598 reads) and (2) a 235 bp fragment on the 5′ region of the COI barcoding region (amplicon COIS 6,440,638 reads), from 32 samples comprising 16 eDNA and 16 invertebrate community DNA samples. Data for these two amplicons were obtained from

a larger data set including additional amplicon libraries, sequenced on two lanes of MiSeq. Overall, the eDNA samples (extracted from filtered water samples) achieved good sequence coverage (mean number of reads per sample ( ± s.d.): COIF: 269,769 ± 57,427; COIS: 259,723 ± 85,437; for exact number of reads per sample, see Supplementary Table 1). Some of the community DNA samples that contained only small amounts of pupal exuviae resulted in a lower number of reads for both amplicons.

**Control samples**. During PCR screening of negative controls, no band (no amplification) was observed on agarose gels. Regardless of no visual proof of amplification, each sample was sequenced and a very low number of reads was returned. After PCR and sequencing of the negative control samples, COIS detected only two OTUs, which were BLAST identified as bacteria. For COIF, again only two OTUs were detected, identified as Gastropoda and Diptera. The Gastropoda OTU was represented by 240 reads in one of the controls while the Dipteran OTU was only represented by 10 reads in total across all types of negative controls.

The positive controls yielded good results for both amplicons, with 547,730 (COIS) and 393,341 (COIF) reads after quality control. Detection success was 100% for COIS (all 30 species detected) and 87% for COIF (26 species detected). Among the species that were not detected was a mayfly species (*E. danica*), which also failed to amplify and sequence during individual barcoding of specimens, using the Folmer primers (Supplementary Table 2). BLAST identification and screening of positive control reads resulted in > 99.9% of the reads being assigned to the target species known to be present in the positive control. The relative abundance of OTUs found in the positive control which were attributed to non-target taxa was 0.026% for the COIS and 0.007% for the COIF (Supplementary Table 3).

**Abundance filtering and rarefaction analysis**. Following investigations of how screening different levels of abundance of rare OTUs affected overall OTU richness (including no filtering, and removal of OTUs that were present at less than 0.01% and 0.02%), a filtering level of 0.01% was set for all ecological analyses. Removal of OTUs present at < 0.01% yielded equivalent levels of OTU genus richness for the community DNA (37 genera) and eDNA (43 genera) according to year 2014 Chironomidae records of Llyn Padarn (31 genera; Fig. 1). Furthermore, filtering of reads below 0.01% was within the limits of a small number of non-target reads detected in the positive control samples. The genus richness comparisons employed COIS data to ensure comparability between eDNA and community DNA for the Chironomidae below. According to the analysis of OTU accumulation curves (Supplementary Figs 1 and 2) versus sequence coverage, a rarefaction depth of 57,869 reads was applied across all water samples (Supplementary Fig. 1a). To subsample Animalia OTUs in our samples a rarefaction depth of 24,914 reads per sample was used (Supplementary Fig. 1b).

**Total taxonomic diversity**. OTU clustering of the combined eDNA and community DNA data sets at 97% similarity cut-off (after removal of low abundance OTUs) yielded: 442 (eDNA) and 309 (community DNA) OTUs for COIF, and 482 (eDNA) and 394 (community DNA) OTUs for COIS. Taxonomic assignment through BLAST identified the majority of OTUs from Animalia and Protista (Supplementary Fig. 3). From the eDNA samples, COIF identified 170 (35.3%) Animalia OTUs, of which 91 comprised Arthropoda (including 42 Insecta), whilst COIS identified 251 Animalia OTUs (56.8%), of which 212 were

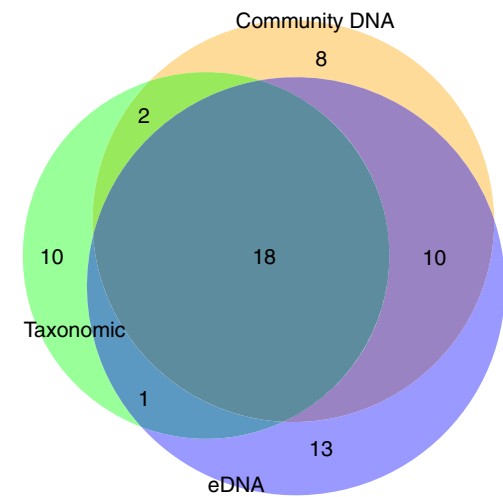

**Figure 1 | Venn diagram showing genera richness detected for all three methods.** Number of Chironomidae genera per sample type (purple, eDNA; orange, community DNA; green, taxonomic identification) and the number of genera common between sample types (overlap area). (Four time points, 45 Chironomidae genera).

Arthropoda (including 167 Insecta; Supplementary Fig. 4). For the community DNA samples, COIF detected 219 (43.6%) Animalia OTUs, of which 171 were Arthropoda (including 132 Insecta), whilst COIS recovered 227 (73.5%) Animalia OTUs, of which 212 consisted of Arthropoda (including 184 Insecta).

Although not the focus of the study, metabarcoding of the eDNA samples (COIS used here as an example) also yielded matches to fish (*Phoxinus phoxinus*), amphibian and terrestrial OTUs represented at high read frequencies or distributed across numerous independent samples. Of the terrestrial taxa, spider OTUs from the Segestriidae (3,753 reads) and Thomisidae (1,858 reads) families, a millipede OTU (7,312 reads), orthopteran OTU (14,237 reads) and 2,114 reads from domesticated cow (*Bos taurus*) were recovered from multiple samples throughout the year, in addition to a broader diversity of terrestrial groups represented at lower frequencies in the dataset.

**Temporal trends of OTU richness from eDNA samples**. Measures of OTU richness were calculated exclusively for eDNA samples and plotted against time to detect possible seasonal variations (Supplementary Fig. 5). All samples were rarefied at an equal depth appropriate for each amplicon (total diversity dataset: 57,869 reads per sample, animal diversity dataset: 24,914 reads per sample, for all water samples).

Mean Animalia richness for COIS ( ± s.d.) was 37.8 ( ± 10.4), and for COIF, 31.4 ( ± 11.4) (Supplementary Fig. 5a). A significant correlation was detected (Spearman's correlation, $P < 0.05$) between the OTU Animalia richness estimates derived from COIF with time and temperature, but not with pH or dissolved oxygen (D.O.). In addition, mean total richness for COIS ( ± s.d.) was 73.1 ( ± 21.2), and for COIF, 88.1 ( ± 26.9; Supplementary Fig. 5b). A significant correlation was detected (Spearman's correlation, $P < 0.05$) between the COIF (total richness), time, temperature and D.O., but not pH. No significant correlation was found for COIS for the Animalia and total richness and any of the above parameters.

**Community structure (β-diversity) from eDNA samples**. We used eDNA samples to look into possible changes in community

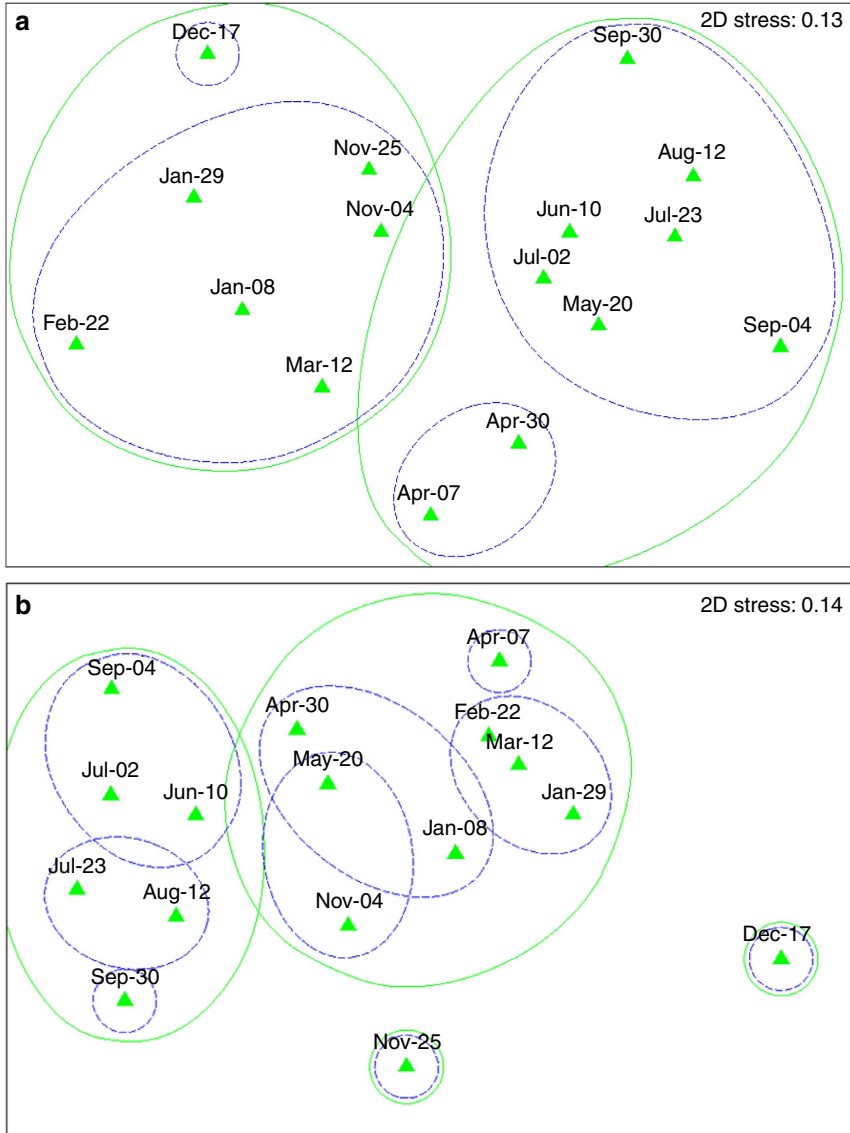

**Figure 2 | Animal eDNA β-diversity–nMDS (Sørensen index).** (**a**) COIF, (**b**) COIS amplicon (eDNA samples only; $N = 32$). Solid green circles: 30% similarity cutoff (corresponding to 'winter'–'summer' groups), dashed blue circles: 40% similarity cutoff.

structure over time, for the Animalia identified diversity as well as the total diversity in the dataset. For the eDNA samples, nMDS analysis (Sørensen index) of total diversity for both amplicons (Fig. 2), delimited patterns of seasonal variations driving community composition with qualitatively higher temporal resolution recovered from the smaller amplicon COIS. ANOSIM analyses also supported two main groupings, 'winter' (November to April) and 'summer' samples (April–Oct) (COIF: ANOSIM sig. level = 0.1%, Global $R = 0.717$, COIS: ANOSIM sig. level = 0.2%, Global $R = 0.475$, with outlying samples from winter sampling). Additional analysis of the total diversity supports similar findings (two main groupings: 'winter' (Nov-April) and 'summer' samples (April–Oct) (COIF: ANOSIM sig. level = 0.1%, Global $R = 0.777$, COIS: ANOSIM sig. level = 0.1%, Global $R = 0.703$)) (Supplementary Fig. 6).

**Temporal trends in Chironomidae richness.** Analyses of untrimmed COIF Chironomidae data suggested that temporal richness patterns between eDNA and community DNA samples

were comparable (Spearman's correlation $P < 0.01$ between eDNA and community DNA for COIF un-trimmed data; Supplementary Fig. 7). Nevertheless, the sequencing coverage of Chironomidae from the eDNA samples were approximately an order of magnitude lower for COIF than for COIS (Supplementary Fig. 2). Subsequently, to maintain a sufficient sequencing depth across samples, COIF was not retained for further Chironomidae related analyses and rarefied incidence based data were used with 4,000 sequencing reads per sample, for COIS only (Supplementary Fig. 2).

For the Chironomidae assigned OTUs, COIS identified 103 OTUs from eDNA and 94 OTUs from community DNA samples (138 unique OTUs in total). Using a combination of BLAST ID $\geq 99\%$ and the online Barcode of Life Database (BOLD) species assignment tool[36], 73 OTUs (53% out of 138 unique) were assigned species level taxonomic information. Analysis of historical species occurrence data collected by the Environment Agency (EA; summer surveys 2003–2013) in Llyn Padarn (N. Wales, UK) indicated the presence of $\geq 99$ Chironomidae species from 57 genera. Moreover, Fig. 1

illustrates the qualitative overlap between the number of chironomid genera delimited by the current community DNA (65%), eDNA (61%) and taxonomy approaches. Similarly, see Supplementary Fig. 8 for overlap between each method for the four summer time points used for analysis.

To visualize the empirically derived annual diversity patterns, OTU and genus richness was assessed against time (Fig. 3) using a polynomial model. Observed OTU richness ranged from 5 to 27 OTUs for eDNA and 1–27 OTUs for community DNA over time (Fig. 3a). Conversely, genus level richness ranged from 5 to 19 for eDNA, 1–16 for community DNA. For the data derived from taxonomic identification of invertebrate (exuviae) community samples, genus level richness ranged from 10 to 18 (green points, restricted to 4 summer sampling times;

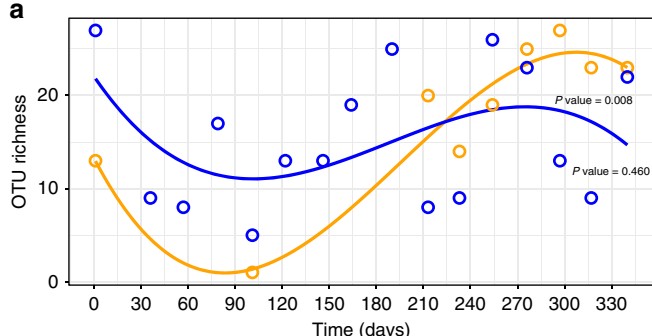

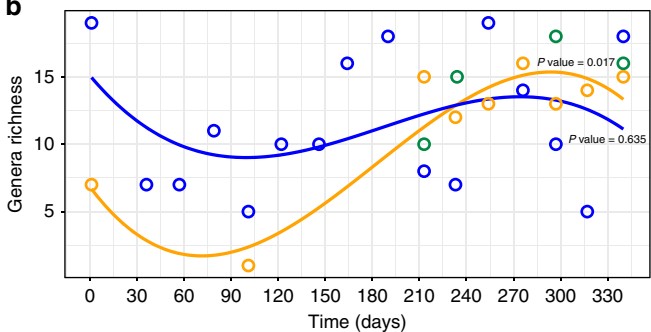

**Figure 3 | Richness patterns for Chironomidae OTUs and genera.**
(**a**) OTU richness. (**b**) Genera richness. Points represent richness values to individual sampling points for eDNA (blue), community DNA (orange) and taxonomic identification of chironomid exuviae (green). Sampling points spanning the winter months (days 36–190) did not yield data due to very low physical numbers of exuviae. Best fitted, significant lines from polynomial regressions for eDNA samples (blue) and community DNA (orange), plotted against time (x axis: September 2013 to September 2014) (**a**) eDNA: P value = 0.46, community DNA: P value = 0.008 (N = 25 datapoints), (**b**) eDNA: P value = 0.017, community DNA P value = 0.635 (N = 29 datapoints).

Fig. 3b). Please also note that sampling points spanning the winter months (days 36–190), which did not yield data, represented samples which contained very low physical numbers of exuviae. Consequently, they were not sequenced to an adequate depth in a mixed Illumina sequencing library, and could not be retained for analysis.

Significant associations were detected between time and Chironomidae OTU and genera richness derived from community DNA (OTU richness: polynomial regression, $R^2 = 0.890$, $P$ value = 0.008; Genera richness: $R^2 = 0.849$, $P$ value = 0.017). However Chironomidae OTU and genera richness derived from eDNA samples did not differ significantly over time (OTU: polynomial regression $R^2 = 0.187$, $P$ value = 0.460; Genera: $R^2 = 0.128$, $P$ value = 0.635; Fig. 3). Taxonomic richness (genus level) also did not differ significantly over the limited time points available from seasonal sampling.

**Temporal variation of OTU Abundance.** We assessed the annual variation in OTU abundance from metabarcoding sequencing reads between eDNA and community DNA sampling methods using a generalized additive model (GAM). To allow across method comparisons we compared OTU abundances for Chironomidae OTUs occurring in both eDNA and community DNA datasets (45 OTUs). Abundances differed significantly among different OTUs (GAM, F = 4.688, P value < 0.001) with a significant effect of the temporal smoothing term (GAM, F = 2.561, P = 0.047; Table 1). In addition, abundances did not differ significantly between methods (GAM, F = 0.013, P value = 0.908), but a significant OTU identity × method interaction (GAM, F = 1.733, P value = 0.003) was found. The abundance of OTU reads was also found to be significantly positively correlated with expected species frequency (ranging from 0.01 to 0.79) across 97 sites in the United Kingdom (UK) (two-way analysis of variance (ANOVA), $R^2 = 0.087$, $P$ value = 0.003; Table 1), using previously catalogued Chironomidae species frequency data[37] (Fig. 4).

## Discussion

We present here one of the first temporal studies of aqueous eDNA and community DNA biodiversity from a lake ecosystem, in addition to targeting a specific group of ecological sentinel macroinvertebrates. In contrast to previous analyses that have used PCR (quantitative PCR) to infer presence/absence of a small number of target species (for example, macroinvertebrates) from eDNA samples[12,13], we employed HTS of amplicon libraries (metabarcoding) to assess temporal trends in total biodiversity. Such methodology allows for the characterization of the entire community, which is not possible through targeted individual-species sequencing that employs taxon-specific primers. Simultaneously, we provide among the first accounts of temporally collected biodiversity data from an annual series of

**Table 1 | Sequence reads versus OTU and sampling method.**

|  | d.f. | F | P value |
|---|---|---|---|
| OTU | 44 | 4.688 | < 0.01 |
| Method | 1 | 0.013 | 0.908 |
| OTU × method | 44 | 1.733 | 0.003 |
|  |  |  |  |
| *Approximate significance of smooth terms* | *e.d.f.* | *F* | *P value* |
| s (time) | 2.899 | 2.561 | 0.047 |

d.f., degrees of freedom; e.d.f., estimated degrees of freedom; GAM, generalized additive model.
Explaining OTU sequence abundance relative to OTU taxonomic ID (OTU) and sampling method (eDNA or community DNA—Method) over time. Model estimates and significances of the smoothing terms are given for the most parsimonious models. ($R^2 = 0.18$).

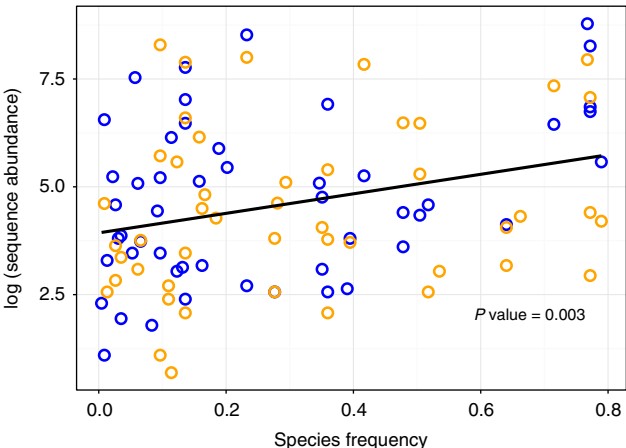

**Figure 4 | Abundance patterns for Chironomidae.** The sequence based Chironomidae OTUs plotted against species frequency across the UK, according to historical data. Environmental DNA (eDNA) samples (blue) and community DNA (orange) are shown along with the best fitted, significant, linear regression model (black line) ($R^2 = 0.087$, $P$ value $= 0.003$, $N = 97$ datapoints).

eDNA samples compared simultaneously with a series of invertebrate community DNA samples. Our findings yield an informative characterization of temperate lake ecosystem-wide biodiversity, through detection of multiple groups of organisms from invertebrates to macro-organisms, of primarily freshwater, but also terrestrial origins. Furthermore, the biodiversity of the indicator taxon group used (Chironomidae) was successfully detected throughout the year, from both eDNA and community DNA samples, exhibiting substantial overlap with traditional taxonomic data. In addition, OTU sequence abundances were significantly positively associated with expected chironomid species abundance based on UK taxa occurrence data (Table 1, Fig. 4). Such direct coincidence, despite potential biotic and abiotic variability in the release, transport and persistence of eDNA[8], demonstrates the value of eDNA metabarcoding for biodiversity characterization and ecosystem monitoring[38].

Both metabarcoding amplicons detected large amounts of Animal phylum level diversity from eDNA samples, showing broad representation across the freshwater taxonomic biosphere, including the Arthropoda (Supplementary Fig. 4). Within the Arthropoda, the dominance of Insecta, Maxillopoda and Malacostraca (Crustacea) also demonstrates the utility of eDNA metabarcoding for characterization of freshwater ecosystem-wide biodiversity. There is increasing exposure of the use of eDNA metabarcoding for the detection of fish and amphibians[19,20], as also recorded here. A more novel concept is the ability of freshwater systems to integrate eDNA biodiversity information from terrestrial sources. Terrestrial species found in our dataset, such as spider, millipede and orthopteran species, or the ubiquitous *Bos taurus* (please also note, that no bovine serum products were used in the HTS library preparations), are all commonplace in the surrounding area of the study site and were detected by the analysis of eDNA residing in the lake water samples. The ability of freshwater catchments to contain eDNA from broader habitat biodiversity therefore presents an opportunity for further research regarding the relationship between aqueous eDNA and biodiversity at the landscape scale.

Focusing on the Chironomidae richness estimates derived from the analysis of the short COI fragment (Fig. 3), we can see that the COIS amplicon yielded 138 unique OTUs from both sample types throughout the year. The analysis of the COIS amplicon therefore provided valuable comparative qualitative and quantitative data both within the metabarcoding datasets and between the historically collected data for Llyn Padarn and the rest of the UK[37]. Other eDNA studies have focused mainly on macro-organisms such as fish or amphibians whereby skin cells and mucus are a likely primary source of eDNA[8]. While aquatic invertebrates such as chironomids are individually typically much smaller, the accumulated biomass of the community clearly produces sufficiently detectable and persistent amounts of eDNA (from natural shedding, moulting and death) for meaningful biodiversity assessment. Additional quantitative studies are required to determine the effects of invertebrate community biomass on levels of eDNA in environmental samples[21].

Sequencing of the complete COI region (COIF $\sim 658$ bp) from eDNA samples was successful in detecting several genera of chironomids and provided biodiversity estimates comparable with community DNA biodiversity patterns (Supplementary Fig. 7). However, it was not possible to retain the COIF locus throughout all analyses after applying strict abundance filtering of OTUs. Low sequence coverage of the COIF for the Chironomidae (primarily in the water eDNA and not the community DNA samples, Supplementary Fig. 2) meant that more robust, ecological comparisons were more effectively achieved using the short eDNA fragment (COIS). Possible reasons for the discrepancies in coverage of the two amplicons could be related to variations in primer specificity, with the COIS primers being more successful than COIF primers in amplifying Chironomidae[39] (please also see the limitations of the Folmer COI barcoding primers for metabarcoding analyses in[40]). Nevertheless, we did not detect substantial phylogenetic biases in OTUs recovered from the two primer pairs (Supplementary Fig. 9) and coverage of the Chironomidae was only depleted in the water eDNA samples for the COIF. Alternatively, the discrepancy in different amplicon success may be due to the reduced availability of longer sized eDNA fragments in a natural ecosystem[30].

After DNA is released into the environment, the degradation process likely begins, breaking down DNA and yielding shorter fragments. It has been shown that $\sim 400$ bp length fragments remain detectable in water for days to weeks[6,11], with the rate of degradation depending upon various biotic and abiotic factors[27]. Overall, smaller fragments degrade slower compared to longer fragments, suggesting an enhanced probability of detection by studies targeting shorter DNA fragments[41]. The present data support the enhanced detection of shorter eDNA fragments, as evidenced by higher sequence coverage of the Chironomidae by the shorter COIS amplicon in the water eDNA samples. Nevertheless, the data additionally show that longer fragments are available at likely lower concentrations in the wild[30] (represented by the COIF amplicon; Supplementary Fig. 2). Using time versus DNA fragmentation as a working hypothesis for eDNA degradation, longer fragments are predicted to represent more recently living cellular material. It is also therefore noteworthy that among the water eDNA analyses, only the biodiversity delimited by the COIF amplicon yielded significant associations with time/temperature (Spearman's correlation, $P < 0.05$; Supplementary Fig. 5), most likely representing more rapid breakdown of longer eDNA fragments in the lake environment. Nevertheless, higher sequence coverage, or methods that preferentially amplify longer amplicons, are needed to enhance amplification probability for potentially smaller concentrations of longer eDNA fragments in natural systems. Such solutions include the combination of multiple primer

pairs[17], or use of taxon-specific/blocking primers. Other suggested strategies for enhancing HTS of eDNA (where concentrations are sufficiently high) involve direct shotgun sequencing or use of capture probes[28,42].

Among the concerns regarding the utility of eDNA to assess biodiversity, is whether or not species detection represents living or recently living organisms, or communities of 'zombie' DNA (that is, historically distant DNA from organisms that previously lived in the ecosystem a substantial time ago)[38]. If eDNA exhibited long persistence times in the wild, temporal patterns of β-diversity would be predicted to be extremely low (that is, non-existent), especially when derived from smaller fragments. However, here we have clearly shown that temporal turnover (β-diversity) was observed for both the animal level (Fig. 2), and total-diversity-derived eDNA biodiversity analysis (Supplementary Fig. 6), including temporal patterns of seasonal biodiversity groupings over the year. Similar temporal results were observed for both amplicons, with the short eDNA amplicon providing higher temporal resolution. Some winter samples (25 November and 17 December) in the COIS nMDS analysis displayed high levels of β-diversity, since they either contained higher richness (Supplementary Fig. 5, days 57 and 79) or additional cohorts of taxa not present in the remaining samples (Supplementary Fig. 4). In the absence of technical artefacts, the additional turnover in β-diversity observed could be the consequence of extreme storm events that coincided with the winter 2013–2014 sampling (http://www.metoffice.gov.uk/), inputting additional allochthonous eDNA from outside the study area. The time points defining the separation of the two main seasonal biodiversity groups were identified over November and late April; times that also correspond to water temperature below 8 °C (winter samples) and above 10 °C (summer samples). Changes in observed community composition (β-diversity) over April and November (Fig. 2, Supplementary Fig. 6) most likely reflect seasonal turnover, possibly attributed to lake inversion effects[43]. It is known that changes in water temperature around these times of the year (Spring and Autumn), can trigger the loss of water column stratification by mixing due to changes in surface water temperature[43]. Collectively, the demonstration of seasonal turnover of lake eDNA β-diversity supports empirical studies using model ecosystems[43]. Previous laboratory and mesocosm studies have demonstrated the short-term temporal decay of eDNA in artificial environments (for example, 2–6 weeks)[8,22,26] and the present data show that the eDNA signal in the wild is of a contemporary nature.

Metabarcoding sequencing of invertebrate communities directly reveals the presence/absence of living, or recently living communities[28]. Hence, the insights provided by community DNA samples here offered an essential benchmark to serve as a proxy for the contemporary invertebrate community. The biodiversity estimates derived from metabarcoding of the community DNA (Fig. 3; Supplementary Fig. 7, orange lines) matched literature-based estimations of seasonal variation of Chironomidae for Northern Hemisphere temperate latitudes[44], with a decrease in species richness over winter (often represented by 'null' samples due to low numbers of collected exuviae) and a summer increase related to rising water temperature (Fig. 3). Since the emergence patterns of Chironomidae through the year are strongly related to changes in temperature and photoperiod[44] (Supplementary Methods), rapid turnover in emerging communities are apparent and can yield biased estimates of ecological status due to short-term shifts of species emergence[45]. One of the advantages of metabarcoding over traditional analysis is the ability to analyse many samples simultaneously, and so using molecular approaches for biodiversity assessment presents the opportunity to intensify ecological assessment and derive greater precision in ecosystem health assessment[3].

The companion analysis of the chironomid eDNA did not follow the expected emergence pattern in richness, despite detecting Chironomidae turnover throughout the year from community DNA samples (Fig. 3). The combination of the β-diversity turnover in eDNA composition (Fig. 2), seasonally fluctuating community DNA richness (Fig. 3, orange lines) and a lack of coherent seasonal shifts in eDNA richness (Fig. 3, blue line) thereby provides an annual model of 'community DNA—eDNA' dynamics. The data thereby suggest that there will likely be a standing resource of eDNA for biodiversity detection in lake ecosystems that experience annual species turnover[43] (Fig. 2). Compositional turnover is thereby expected to result from seasonal variation in species abundances, increasing sources of contemporary eDNA, and environmental degradation decreasing levels of past eDNA accumulation.

Using GAM modelling facilitated comparison between read abundances of individual OTUs derived from eDNA and community DNA analyses. Numbers of read abundances differed between OTUs over time and between eDNA and community DNA abundances at the individual OTU level (Table 1). There was also a significant positive association between the abundance of sequencing reads derived from the present study and species frequency at the national scale (Fig. 4). Therefore, lower abundance OTUs from the present study occur at lower frequencies and more abundant OTUs are more common, according to an extensive database of Chironomidae occurrence across the UK[37] (Fig. 4).

In combination, the analyses provide an overview of chironomid lake eDNA dynamics. Some species will inevitably yield higher levels of eDNA than others, in relation to life history stage, moulting rates/frequency, abundance, biomass, or cellular content/mitochondrial densities[3,7,8]. In addition, the relationship between eDNA and community DNA is affected by biophysical characteristics and interactions between biotic and abiotic factors (for example, microbial activity, UV radiation and temperature) that affect persistence and degradation rates throughout the year[8,27]. Despite such dynamic interactions, numerous broad quantitative associations have been reported for a range of taxa and their eDNA profiles, including data from artificial, semi-natural and natural aquatic ecosystems[46–51]. Here also, regardless of which methodology was employed, metabarcoding of both eDNA and community DNA reflected general Chironomidae species frequencies across the UK[37] (Fig. 4) and overlapped with biodiversity estimates derived from taxonomic analyses (Fig. 1).

In summary, we have shown that eDNA from water samples collected consecutively over an annual cycle in a lake ecosystem reveals ecologically representative species and community-level shifts in diversity. Importantly, such patterns were validated both by independent assessments of changes in physical presence in a key indicator group of macroinvertebrates, as well as coinciding with established seasonal trends in indicator species emergence and traditional taxonomy. Collectively, the findings address key outstanding questions related to the ecological relevance and temporal persistence of freshwater eDNA in a natural ecosystem, with significant implications for biomonitoring and the future investigation of biodiversity ecosystem function relationships.

## Methods

**Field sampling.** Samples (chironomid pupal exuviae and water samples) were collected during September 2013 to September 2014 from Llyn Padarn, UK (Supplementary Methods), an oligotrophic lake ecosystem located in Snowdonia

National Park (53.130051, −4.135567), N. Wales, UK (Supplementary Fig. 10; Approximate surface area 97.6 ha, maximum depth 27 m). The site has been monitored regularly by the UK Environment Agency (EA), and more recently by Natural Resources Wales (NRW) for indicator species of Chironomidae and other invertebrate communities, providing important historical data. Two sites at opposite sides of the lake were selected for sampling: Site 1 (S1: NW: 53.139106, −4.153975) and Site 2 (S2: SW: 53.122414, −4.126761) (Supplementary Fig. 10). Using two locations increases potential for species detection based on both eDNA and invertebrate sampling. Sampling was conducted at approximately 3-week intervals for 1 year (16 time points), using standardized sampling methodology, and collecting simultaneously water and Chironomidae samples. The two sites were sampled always in the same sequence (S1, then S2) between 08:30–11:30 hours, including consecutive collection of water samples, invertebrate samples, followed by water metadata (pH, Dissolved Oxygen (D.O.), conductivity and water temperature), using a calibrated YSI Pro Plus multi-meter. As only water and exuviae (shed skins) were collected and the work was performed in collaboration with the EA and NRW, a permit was not required.

**Chironomid exuviae collection and eDNA filtration.** Invertebrate samples in the form of chironomid exuviae (shed pupal skins) were collected using the field collection protocols for the Chironomid Pupal Exuviae Technique (CPET)[52], with a 250 μm mesh collection net (Supplementary Methods). The floating insect skins were collected on the leeward side (accumulation area) of each sampling site following described methods[34] and placed in a sterile container. Upon returning to the lab, the sample was coarsely sorted to remove excessive plant debris, fixed in 100% ethanol and stored at 4 °C on the same day of collection, until further processing.

For eDNA samples, 1 l of surface water was collected using sterile glass Nalgene bottles from each site, which was transferred on ice and placed at 4 °C immediately after return to the laboratory. Filtration was completed within 6 h in a PCR-free separate room. Sterilized, reusable funnel filtration units (Nalgene filter holders with funnel) were used with 0.45 μm cellulose nitrate filter membranes and a high-pressure vacuum pump. The filter membranes were stored in sterile 15 ml falcon tubes at −80 °C until DNA extraction.

**Equipment sterilization and negative control samples.** All equipment was thoroughly sterilized between sampling visits. The glass Nalgene bottles used for water collection, filtration units and forceps would undergo consecutive cleaning rounds including wash and overnight soak with 10% Trigene (Ammonium chloride & hydrochloride, Medichem Int.), thorough rinse, UV treatment for 5 min and autoclaving. All additional equipment used for invertebrate collection (net, meters and boots) was also thoroughly washed with 10% Trigene. For eDNA extractions, single-use pre-sterilized scissors and forceps were used to handle the filter membranes, and the exterior of storage tubes was wiped with 10% Trigene before handling. During field surveys, to minimize cross contamination from consecutive sampling points, the water samples were collected first, before any other samples or measurements were taken and prior to invertebrate collection. Negative controls were collected by filtration of 1 l of distilled water through the filtration funnels and filter membranes processed. Blank extractions of reagents (reagent controls) and filters (filter controls) were also extracted with the same Phenol Chloroform extraction protocol (PCI)[53]. The negative-control equipment would undergo the same cleaning steps as stated above. All negative controls were amplified with both primer pairs and MiSeq library preparation steps (as below), and sequenced on an Illumina MiSeq.

**DNA extractions for eDNA filter membranes and invertebrate samples.** Environmental DNA (eDNA) was extracted from the filter membranes, using a modified Phenol Chloroform protocol (PCI), adapted from Renshaw et al.[53], with an added digestion step, with the addition of 20 μl Proteinase K (20 mg μl⁻¹; Sigma-Aldrich) and incubation at 60 °C for 1 h. This protocol was selected after rigorous in-house testing of available eDNA capture and extraction protocols (Supplementary Methods). In Renshaw et al.[53] it was demonstrated that the latter protocol yielded the highest number of DNA copies of targeted eDNA fragments. Furthermore, the combination of filtration and PCI has been shown to optimize DNA yields, performing equally well in eukaryotes and prokaryotes, with enhanced detection of diversity than other methods[54]. Two individual extractions were performed for each sample, which were subsequently pooled. Extractions were performed in a different building to PCR library construction where no invertebrate DNA had been handled previously. Extracts were stored in a clean room with no post PCR processing.

DNA extraction from the bulk pupal exuviae samples (community DNA) was performed using a modified QIAmp Blood Maxi Kit protocol. Due to seasonal variation of chironomid emergence[44], the mass of the collected invertebrate skin material varied, with some of the winter samples containing smaller amounts of tissue. To optimize extraction efficiency, 1 g of dry invertebrate material was subsampled from large samples. Conversely, for some low-density winter samples, 1 g of exuviae was not available and so in these instances, the whole sample was used for analysis. DNA extraction was performed in standard Qiagen Blood and Tissue kit columns for small winter samples and QIAmp Blood Maxi Kit columns

for all other samples with an added 20 μl Proteinase K (20 mg μl⁻¹) overnight incubation step. Both kits are verified by Qiagen to use the same chemistry and differ with respect to the use of columns of different volume capacity to prevent clogging of the membrane. Following separation from the ethanol preservative, the community samples were allowed to air-dry for ~1 h and then were homogenized using a sterile mechanical drill and pestle. For detailed information on each extracted sample, see Supplementary Tables 4 and 5.

**Primer selection and MiSeq Library preparation.** To fulfil the overarching aims of the study, we required (a) metabarcoding primers that would amplify across a broad range of taxa (in particular, lake occurring taxa), (b) a marker enabling the best annotation power for macroinvertebrates and in particular, the Chironomidae, (c) a combination of two primer pairs providing different length amplicons.

Accordingly, two amplicons of different sizes of the mitochondrial Cytochrome Oxidase I gene (COI) were selected for sequencing. The full-length COI barcoding region (658 bp), using the universal Folmer primers LCO1490 - HCO2198 (ref. 55) (amplicon COIS) and a 235 bp fragment (amplicon COIS) using the forward primer LCO1490 and the reverse COIA-R primer (reversed forward COI-A primer by ref. 39; see Supplementary Table 6 for primer sequences). Initially, the forward COI-A primer was designed by ref. 39 specifically for amplification of Chironomidae from environmental samples. Two Illumina MiSeq dual indexed amplicon libraries were prepared using a two-step PCR protocol[56]. The first round amplification was performed using template-specific primers with 5′ Illumina tails (TruGrade, by IDT, Integrated DNA Technologies (Coralville, USA)), followed by Agencourt AMPure magnetic bead purification. A second round amplification was performed using Illumina adapters with eight-nucleotide Nextera indexes (Supplementary Table 6). A 5N sequence was implemented between the forward universal tail and the template-specific primer, which is known to improve clustering and cluster detection on MiSeq sequencing platforms[56]. Using primers with identical tails in the first step and indexed primers in the second, is a protocol specifically developed by Illumina to reduce bias caused by variable index sequences in mixed environmental samples[57,58].

Each sample was amplified in triplicate, the final products were pooled and purified with AMPure beads and quantified using a dsQubit assay. Final library pooling was performed in equimolar quantities for all samples. Sequencing was performed at the Liverpool Centre for Genome Research, distributed across two independent lanes (for the COIS and COIF amplicons) of Paired-end Illumina MiSeq (2 × 300) sequencing.

**PCR protocols for MiSeq library preparation.** PCRs were performed in 25 μl reaction volumes containing, for Round 1: 12.5 μl Q5 Hot Start High-Fidelity 2X Master Mix, 10.5 μl PCR water, 0.5 μl (10 nmol μl⁻¹) of each forward and reverse primer and 1 μl DNA (10 ng μl⁻¹). For Round 2: 12.5 μl Q5 Hot Start High-Fidelity 2X Master Mix, 6.5 μl PCR water, 0.5 μl of each forward and reverse primer and 5 μl Purified PCR product from Round 1. The following thermo-cycling parameters were used: Round 1: COIF: Denaturation at 98 °C for 30 s, 20 cycles of: 98 °C for 10 s, 46 °C for 30 s, 72 °C for 40 s, followed by a 10 min extension at 72 °C, hold at 4 °C. COIS: Denaturation at 98 °C for 30 s, 20 cycles of: 98 °C for 10 s, 45 °C for 30 s, 72 °C 30 s, followed by a 10 min extension at 72 °C, hold at 4 °C. Round 2: both amplicons: Denaturation at 98 °C for 30 s, 15 cycles of: 98 °C for 10 s, 55 °C for 30 s, 72 °C for 30 s, followed by a 10 min extension at 72 °C, cool at 4 °C for 10 min. Round 1 PCRs were performed using Illumina-tailed primers and Round 2 using Illumina indexes.

**Positive control samples.** To account for efficiency of amplification protocols and sequencing, a composite positive control sample comprising 30 invertebrate DNA extracts, including Amphipoda, Coleoptera, Diptera, Ephemeroptera, Gastropoda, Hemiptera, Isopoda and Trichoptera, was also amplified in triplicate with both primer pairs, and sequenced alongside eDNA and community samples on MiSeq (Supplementary Table 2).

**Bioinformatics and statistical analysis.** Sequences, including positive and negative controls, were de-multiplexed and Illumina adapters trimmed using Cutadapt[59] and Sickle[60]. A 10% level of mismatch (2 bases) was allowed for primer removal. Filtering and quality control were then performed using USEARCH v7 (ref. 61). Sequence quality was visualized using FastQC (www.bioinformatics.babraham.ac.uk) and only sequences with a Phred quality score >25 were retained for analysis. Using USEARCH (fastq_maxee=1) sequences with a maximum expected error (maxee)>1 were discarded. Maxee is the expected number of errors as sum of the error probabilities (provided by Phred scores). Filtering was performed after merging of R1 and R2 reads (minimum overlap 25 bp), which allows recalculation of the error probabilities for the combined sequences and increased accuracy. Sequences shorter than 100 bp were discarded. The remaining sequences were de-replicated and sorted by cluster size (cluster abundance) and sequences with <2 clusters (singletons) were removed. For the COIF amplicon, the whole barcoding region was amplified and sequenced, but because of the current limitations of MiSeq sequencing read lengths, only the forward reads (R1) were used for analysis. Consequently, the per

base quality drop expected in Illumina MiSeq data at the tail of the forward reads was inspected in FastQC and all reads were truncated at 250 bp and then quality filtered as above. Next, chimeras were removed (uchime_denovo) using a de novo delimitation approach. An operational taxonomic unit (OTU) table was created using OTU clustering at 97% similarity (USEARCH). Clustering at 97% similarity level was chosen based on existing knowledge of intraspecific diversity for Chironomidae[39], since previous studies suggest that chironomid intraspecific diversity ranges between 0 and 4.2% (ref. 39) or 0 and 4.9% (ref. 62).

Taxonomy was then assigned to the OTU table using BLAST+ (megablast)[63] against a reference COI database. The reference library was compiled from NCBI GenBank, by downloading all COI sequences, >100 bp, excluding environmental sequences (20 June 2015, N = 807,388 sequences) and higher taxonomic level information was edited using the GALAXY online software platform[64]. Taxonomic assignment of the OTU tables and subsequent analysis was performed in QIIME[65]. All analyses involving USEARCH, QIIME and BLAST+ were performed using the High Performance Computing Wales systems.

Given the potentially sensitive nature of eDNA metabarcoding, low frequency sequences can either represent less abundant taxa, or possible false positives and low-level contaminant OTUs[66]. To reduce the error associated with low frequency sequences, and also focus analyses on predicted levels of richness[67], we used two types of analysis. First, we identified the frequency of potential contaminant reads in the positive control. Second, we compared chironomid eDNA richness with variable levels of relative abundance filtering (no filtering, 0.01 and 0.02%), against historical records of richness (genus level only available) for Llyn Padarn (based on summer surveys for Llyn Padarn, 2003–2013). Consequently, abundance filtering was performed on the OTU tables at the level that most closely emulated expected chironomid richness and within the limits associated with empirically observed low-level contamination in the sequencing dataset.

The validity of the Chironomidae OTUs identified by BLAST and retained after abundance filtering was checked using a phylogenetic approach. The BLAST identified Chironomidae OTUs were aligned with barcodes from 24 Chironomidae and 40 Trichoptera species obtained herein, sequenced from UK samples using universal primers[55]. Alignment, testing for the presence of stop codons, insertion/deletion events and bootstrapped phylogenetic tree construction were performed in MEGA[68]. Ultimately, only the OTUs that grouped closely with known chironomid sequences on the phylogenetic tree were included in further analysis.

For downstream analyses, the appropriate depth of coverage per sample was determined according to OTU accumulation versus sequence coverage curves generated in QIIME. Samples were subsequently normalized using rarefaction in QIIME at appropriate depth for each amplicon[69].

**Taxonomic identification of invertebrate community samples.** To provide a comparison with community DNA and eDNA sequenced samples, chironomid exuviae community samples from 4 time points (T10: 30 April, T11: 20 May, T14: 23 July, T16: 4 September) were taxonomically identified according to standard CPET methodology used by the EA. More specifically, 200 chironomid exuviae were subsampled from the total community sample and identified to the highest possible level (genus or species) by specialized EA staff. The results of the taxonomic identification were used to compare chironomid richness at the genus level with metabarcoding-generated richness (see below).

**Calculation of diversity measures.** OTU richness (total diversity and Chironomidae diversity) was calculated in QIIME. Furthermore, for Chironomidae with good taxonomic identification, richness was also calculated at the genus level. To assess variation of richness over time polynomial regression was performed using R version 3.2.4 (2016).

The PRIMER-E software[70] was used to calculate β-diversity based on the Sørensen index for total diversity and Animalia only diversity detected from aqueous eDNA samples and for Chironomidae OTUs for both sample types. Non-metric multi-dimensional scaling (nMDS) and Hierarchical Clustering (HC) analysis were used to represent community similarity between samples. Analysis of similarity (ANOSIM) was used to test for significant effects of time in relation to community composition.

**Chironomidae OTU read abundance (eDNA versus community DNA).** To explore relationships between the numbers of metabarcoding sequence reads, individual OTUs and methodology (eDNA versus community DNA), we used a generalized additive model (GAM), with time as a smoothing term, using the R-package mgcv[71]. In the GAM model, abundance, calculated as total normalized reads per OTU and standardized per method (to allow for across method comparison), was assessed in relation to OTU identity and method (eDNA versus community DNA). In addition, we assessed the ecological relationship between OTU abundance (log transformed) in Llyn Padarn and species frequency (that is, abundances derived from ecological assessment) across the UK, by performing a two-way ANOVA, using the lm function

in R. UK species frequencies were derived from a Chironomidae inventory of 435 species across 220 UK lakes[37]. We restricted the species frequency data to 97 sites where species frequency was inventoried at the national level and observed in this study.

**Data availability.** Sequencing data reported here have been deposited in GenBank (accession numbers: KY225332 - KY225378 and KY225379 - KY225480) and the European Nucleotide Archive (ENA) (accession number: PRJEB13009).

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

## Acknowledgements

This work was funded by the Environment Agency (EA) UK, a Knowledge Economy Skills Scholarship (KESS), a Natural Environment Research Council (NERC) NBAF pilot project grant (NBAF824 2013–14) and the Freshwater Biological Association (FBA; Gilson Le Cren Memorial Award 2014). We thank the EA and Bangor University for support and in particular, Wendy Grail, John Evans, Emlyn Roberts and EA staff for facilitating provision of eDNA grade laboratory working spaces, equipment, and taxonomic identification of chironomid specimens; High Performance Computing Wales for allowing use of their systems; Les Ruse and APEM for identification of Chironomidae specimens for Barcoding; Natural Resources Wales for providing historical data for Llyn Padarn. We also acknowledge the support of NERC Highlight Topic grant NE/N006216/1. Knowledge Economy Skills Scholarships (KESS) is a pan-Wales higher-level skills initiative led by Bangor University on behalf of the HE sector in Wales. It is part funded by the Welsh Government's European Social Fund (ESF) convergence programme for West Wales and the Valleys.

## Author contributions

I.B., S.C., G.R.C.: designed experiment. I.B.: performed lab work, fieldwork, bioinformatics and statistical analysis. M.S.: performed statistical analysis and data modelling, D.L.: contributed in optimization of analytical pipelines. M.H., M.C., K.W.: participated in experimental design. I.B., S.C., G.R.C.: wrote manuscript. I.B., S.C., G.R.C., M.S., K.W., D.L., and M.H.: edited manuscript.

## Additional information

**Competing financial interests:** The authors declare no competing financial interests.

