## [Peer Review File · Nature Communications]

Reviewers' comments:

Reviewer #1 (Remarks to the Author):

The study uses PCR and sequencing on water samples from lakes, i.e. 'free' DNA, to determine the composition of animal communities and their seasonal turnover. The study is mainly methodological, although it attempts to draw broader ecological conclusions about the lake ecosystems. This type of paper is needed to establish the procedures for biomonitoring using novel DNA techniques. Currently increasing numbers of these papers are presented in specialist journals like *Molecular Ecology Resources* and *Methods in Ecology and Evolution*, but the authors seem to believe that this study is of more general interest, mainly because of their findings of temporal turnover that can be detected with these new techniques.

The main attraction of the paper is a demonstration that it is possible to detect a huge diversity of species directly from a water sample that is filtered through a nitrocellulose filter. This is compared to a sample from a 'community sample' that extracts DNA directly from specimens (or dead exuviae of chironomids; this isn't terribly clear). Amplification with primers producing a long (650 bp) and short (250 bp) fragment then should produce a mixed assemblage of PCR products, but less efficiently with the long fragment in the more highly degraded eDNA sample. The authors find well over 2000 species from 12-15 animal phyla in all of these samples, which seems rather phenomenal, but probably in line with other studies of metabarcoding, which now use a standard approach based on the Usearch pipeline for these purposes.

The paper seems rather difficult to read because of the poor structure, even if reading the Methods first (which are appended to the end). I was struggling in particular about how the distinction is made between the 'eDNA' and 'community' samples. The first is by filtration of the water, the second by scooping off the exuviae of chironomids from the surface, if I understand correctly. In both cases huge numbers of species are recovered, presumably because in both cases small individuals including microbes are filtered out of the water. Hence the DNA is not 'free' in most cases. This is perhaps not terribly important, if the aim is to gather the species diversity of a lake from a few litres of water, but it's not 'eDNA' in the strict sense. The authors made no attempt to characterise their samples, not even for the most basic parameters such as DNA concentration and average fragment length. The degree of degradation of the two samples is characterised by comparison of the short and long PCR, but this test is compromised by the selection of different primer sequences in one of the two primers, to include a more taxon-specific sequence.

The analysis is then focused extensively on the species richness and turnover with the seasons, resulting in the intuitive result that there are slightly more species in the warmer months. I don't think this is a big deal and should be toned down. The more important question is about the identity of the various chironomid species, and how they individually respond, in particular in relation to their close relatives (on the tree built from these data). There is the potential for lots of interesting analyses, but we are left with a very simple tree that mainly illustrates the presence/absence of particular species in the two samples. It is

not linked to known species, although it would be very good to establish if the species detected here are indeed those known from the lake. This leads to the wider question about the identity of the ~2000 species found, whose identification greatly depends on existing public databases. What is the confidence? The reader wants to be convinced that this massive sequence variation is a meaningful description of the life in the pond.

My suggestion is to slim down this paper mainly by removing the somewhat repetitive and unconvincing analysis of temporal turnover and to focus more on the methodological aspects of community characterisation. How much of the eDNA is derived from living things, and is it really possible to do barcoding on free DNA? How much 'false positives' are obtained by the Usearch process? How secure is the identification, and do these two different primer sets arrive at the same identification? In addition, rather than drawing the NMDS plots that are very popular in metabarcoding studies but ultimately rather uninformative, perhaps a focus on particular species and their turnover could be more interesting. I don't think the power of metabarcoding is primarily in the quick-and-dirty plot of species diversity (even if that was ok for the early papers in the field demonstrating the feasibility of metabarcoding in principle). This would fall short of conventional studies in community ecology that rely on proper identification and evolutionary analysis of the components.

Reviewer #2 (Remarks to the Author):

A. The authors applied eDNA metabarcoding from lake water samples to screen for changes in diversity over the course of one year. Focus was on chironomid diversity as well as of the lentic community. Furthermore, the authors used two different COI primer combinations (658 bp vs. 250 bp) to test for their efficiency in metabarcoding. The data reveal an enormous diversity of OTUs / taxon richness and using the data the author show clear annual changes in biodiversity patterns and thus the power of eDNA as a tool for biodiversity assessment. I enjoyed reading the manuscript but have a few major points to raise concerning data analysis and interpretation (see C-F).

B. High! Great interest to broad readership!

C. Sampling, laboratory and bioinformatic methods mostly sound but still with few major issues and at least missing important information at several parts.

- Was it always the same person performing sampling, extraction and PCR?
- Please elaborate further on standards (The room where water was filtered was PCR-free, but what kind of other samples are handled in this room? Was the room prepared for eDNA extractions, e.g. by filtering air and UV treatment? Same for the building where extractions were performed; where protective clothes used). Since you used reusable filter funnel units, how did you make sure that they were not contaminated when used for the second time?
- Please add information on how you handled clogging filters (we experience this issue). How many filters were used per sample?
- "For large samples, ~1g of dry invertebrate material was subsampled from the bulk, whereas for some low-density winter samples, the entire sample (<1g)".

By using different amounts of tissue, you possibly introduced bias due to oversampling some and undersampling other samples. Why did you not use the same amount of tissue (<1g) for all samples?

- Also: Please state that a sampling permission was obtained from Snowdonia National Park officials.

- Each sample was amplified in triplicate. Please provide some more details on library preparation and how individual samples were tagged (issues of amplification bias due to indexes possible?).

- It is very important to have a table that lists read numbers per replicate. Please include these information!

- Please explain why only singletons were removed and not reads up to a certain (low) threshold in order to avoid biodiversity overestimation due to reads that are PCR artifacts.

- While all tools used represent absolutely high-end tools for bioinformatic analysis, there are some issues that should be changed. Clustering was performed independently for the two read data sets. However, for a much better comparison the authors should do clustering together for both data sets (COIF, COIS).

- When comparing efficiency of the long Folmer fragment (watch out, mistakenly it is written HCO2189 instead of HCO2198 at one point) to the short fragment the improved efficiency cannot solely be assigned to the length. The primer used (from the Carew et al. study) has a degenerate base ("H") at the very last bases (not present in the Folmer primers). Thus, improved efficiency can be a result of this! I would definitely expect more OTUs through this approach.

- Please use a full stop and not a comma as a decimal separator in Fig. 1.

D. Mostly correct.

But:

- While it is useful apply rarefaction, limiting analysis due to the lowest read number (476 reads) is a concern. I would advise to analyze the data not including the sample with such a low read number to test if results differ. 476 reads (COIF) is by far too low for an NGS OTU analysis.

- I suggest using not the Phred Quality Scores but the USEARCH scores for quality trimming.

- Page 12, line 7: "Both amplicons presented the same annual pattern, with a decrease in species richness over winter and a summer increase related to rising water temperature (Fig. 3a-b, orange lines)." This is not correct. The orange line clearly does not support a clear pattern of increase during the summer months and decrease during winter months: for COIF, community OTU richness is high in January, even higher than in some summer months. For COIS, there are just two data points for community OTU richness between September and May, which does not allow inferring clear patterns from the data. Thus, I suggest to refrain from drawing best fit curves when the data do not permit it and rephrase the sentence accordingly. For eDNA (blue line and dots), there is a gap of data points between February and June in the COIF dataset. Again, it is not advisable to draw a best fit line. Also, the fact that these data points are missing is not discussed in the manuscript (they are present in Fig S3). The emergence pattern of chironomids can hardly explain the complete absence of any DNA in the water and rarefaction analyses should also not cause the observed pattern, especially since lots of OTU were present in February. Please discuss

possible reasons for the lack of data.

E. Part of the conclusions are correct but mainly available data should be included (tables on libraries / read numbers / gram of tissue for bulk samples), clustering be performed together for consistency (COIS and COIF), possibly use a lower threshold to avoid including OTUs that are PCR artifacts, avoid including the n=476 samples for rarefaction and reanalyze. Explain missing values in the time series and more carefully interpret the fitting lines of the model due to the missing data points (see comments above). No new experiments needed, no other software approaches needed.

G: Appropriate

H: Abstract, Introduction and Conclusion extremely well written and clear!

Minor points:

Introduction:

"eDNA ecology" is not the right term, please consider rephrasing the sentence. E.g. "...several factors of eDNA research..."

Discussion:

Page 10, line 15: " In contrast to previous analyses of macroinvertebrate species from eDNA samples 19,20, we employed a metabarcoding approach instead of PCR-based detection". This is not entirely correct, since metabarcoding is still a PCR-based approach. Please remove this sentence or rephrase it accordingly.

Page 12, line 1: " Other possible approaches for HTS of eDNA are direct shotgun sequencing of DNA extracts or use of capture probes". While the mentioned techniques are very promising without doubt, please keep in mind that the amount of DNA needed for direct HTS sequencing is relatively high and cannot be easily obtained by sampling few ml or l of water. Please rephrase this sentence accordingly.

Page 8: Dissolved Oxygen "D.O."

In general: When using the scientific name "Chironomidae" there is a capital C. If you write about chironomid samples, it is a small c.

Reviewer #3 (Remarks to the Author):

Bista et al. "Annual time-series analysis of aqueous eDNA reveals ecologically relevant dynamics of lake ecosystem biodiversity".

The authors investigate the use of eDNA metabarcoding of lake communities in parallel with bulk sample metabarcoding of Chironomids. They do temporal analyses of 16x2 samples

taken over 1 year. Although the approach of temporal assessment of diversity by eDNA is fairly new, Biggs et al. (2015). *Biol. Cons.* analysed newt eDNA collected 4 times during the breeding season. The results of Bista et al. are interesting and the paper is generally very well written, but there are some major considerations to be dealt with.

MAJOR COMMENTS

1. The consideration and careful trimming for PCR and sequencing errors
2. PCR tagging system in two rounds of PCR
3. Choice of target region (COI)

1. The consideration and careful trimming for PCR and sequencing errors.

It is well known that numerous errors occur during PCR and sequencing (Schnell et al. 2015. *Mol. Ecol. Res.*, Coissac et al. 2012. *Mol. Ecol.*).

The authors do not describe adequately how errors were dealt with besides that chimeras were removed. The fact that they retrieve "an impressive amount of diversity from eDNA samples" and "with up to 2730 OTUs from one amplicon" should cause some critical consideration. I do not think it is unlikely that high diversity of invertebrates could be obtained by eDNA, but one must be cautious. Especially when the numbers is this high in a northern European lake in one liter of water!. Fx. the authors retrieve taxa in high numbers that are very rare in lakes (Arachnida!, besides the few species such as *Dolomedes* and *Argyronecta*). One approach is to check if these are variants of the very abundant insect reads. It would be very obvious to provide a full species list (especially for the COIF region that can go to species level) to see if/how many "exotic" species turn up. This can be used to evaluate the amount of recovered MOTUs compared to known species diversity in the mock sample. Is significantly more/less MOTUs/unique sequences being recovered than there are "species" added in the positive control sample? Another obvious approach is to look into the non-target reads of the positive controls and use this as trimming. Some similar approaches are in Port et al. (2016). *Mol. Ecol.*

I do realize that the authors find separation of season in the NMDS analyses and that such an overall ecological signal could be retained regardless of stringent trimming, but errors still needs to be taken into account, and how this influences the results.

2. PCR tagging system in two rounds of PCR.

As I understand, the authors use a first round of PCR with the Illumina adaptor and a second round with individual indexes? This approach could cause problems since the individual PCRs (samples and replicates) are not tagged from the first round of PCR. This can severely increase cross-contamination risk if done in parallel for this many samples, and if so, there is no way of checking from which sample a sequence originally came from. See also DeBarba et al. 2014. *Mol. Ecol. Res.*

3. Choice of target region (COI).

The use of COI for DNA metabarcoding studies have been criticized (Deagle et al. 2014. *Ecol. Letters*). The primers are not very universal when it comes to equal amplification chance of all taxa, which can skew the final results on community composition and thus the

interpretations of these results. The authors need to address this in the discussion and do some in silico testing of which taxa will be over/under represented in PCRs and thus in the final sequence reads.

MINOR COMMENTS

#Abstract:

The first sentence is not very clear. The "sensitivity" of what? Also "free" is not a good word as you don't know if the eDNA is in fact extracellular.

What is meant by "efficacy of the ecological signal"?

"for elucidating drivers of biodiversity change over time" should be changed to "for tracking biodiversity change over time". Think this is more correct

"enhancing efficiency"? of what?

#Introduction:

"it is essential that we employ new, reliable and cost effective methods for biodiversity assessment". Why is "new" relevant here?.

"shed skin cells, urine, faeces or saliva of taxa". Taxa should be changed to organisms.

#Results:

Define COIF and COIS the first time (Full and Short)

"mean number of reads {plus minus}SD: COIF: 269,769 {plus minus}57,427; COIS: 259,723 {plus minus} 85,437". Mean number per sample? Please specify!

"393(COIF) and 547 (COIS)". Write the full number (393.000 etc.) to be consistent with above text.

"cubic model explained the highest proportion of diversity in our data". Don't you mean that it explained the variation in our data?

"showing ecologically meaningful sub-groupings within "winter" and "summer" samples". Please explain and expand on how this was meaningful.

"OTU richness for both amplicons ($p < 0.05$), and between D.O. and community DNA". What is D.O??

"throughout the Chironomid tree of life". Would use another word, fx. across the Chironomidae

#Discussion:

"Our findings yield a clear characterisation of temperate lake ecosystem-wide biodiversity".
Please specify why this is so?

"with likely co-variance with productivity". Specify productivity! Is it primary productivity?
"Evidence supports the preferred use of longer amplicons to maximise clear and temporally short-term shifts in community diversity". I think you mean that your results (in stead of "evidence") support the preferred use of.....

"Variations in community composition (β -diversity) that are observed over April and November could either reflect seasonal turnover or possibly be attributed to lake inversion effects". I guess lake inversions would in fact be characterized as one of the drivers of seasonal turnover.

#Methods:

"it was demonstrated that the latter protocol yielded the highest number of DNA copies".
For what species/taxa??

"Such a strategy ensured that the analyses featured the maximum number of independent samples, covered with an appropriate depth of high quality sequences according to their predicted biodiversity." I cannot see how this is obtained by rarefaction/normalization?

#Figures:

Figure 1. The colours are confusing. Use the same colours for the same designations.
Figure 4a. Only one sample from community DNA in winter ??

#Supplementary Material:

The "CPET technique" is not defined

"10% Trigene" is not defined

"Possibility for collection of larger and more ecologically relevant water sample (2L)" This lacks support in the literature I think.

"Optimal pore size for collection of smaller DNA molecules and microorganisms" But you are not targeting microbes.

Figure S1: What is "low numbers of OTUs in the dataset" ??

Figure S2: Needs scale

Reviewers' comments:

Reviewer #1 (Remarks to the Author):

The study uses PCR and sequencing on water samples from lakes, i.e. 'free' DNA, to determine the composition of animal communities and their seasonal turnover. The study is mainly methodological, although it attempts to draw broader ecological conclusions about the lake ecosystems. This type of paper is needed to establish the procedures for biomonitoring using novel DNA techniques. Currently increasing numbers of these papers are presented in specialist journals like *Molecular Ecology Resources* and *Methods in Ecology and Evolution*, but the authors seem to believe that this study is of more general interest, mainly because of their findings of temporal turnover that can be detected with these new techniques.

1. >>> Indeed, understanding the temporal dynamics of macrobial taxon eDNA in the wild has been highlighted as an outstanding research priority in recent review papers to inform our knowledge regarding the relationship between extant biodiversity and environmental DNA ¹⁻³.

The main attraction of the paper is a demonstration that it is possible to detect a huge diversity of species directly from a water sample that is filtered through a nitrocellulose filter. This is compared to a sample from a 'community sample' that extracts DNA directly from specimens (or dead exuviae of chironomids; this isn't terribly clear).

2. >>> The collection of exuviae samples is presented we hope clearly in the text. Please see Methods: Page 20, Lines 430-432 – Chironomid Exuviae Collection (Invertebrate community samples) and water filtration (eDNA samples)” and Introduction Page 6, Lines 100-109.

Amplification with primers producing a long (650bp) and short (250bp) fragment then should produce a mixed assemblage of PCR products, but less efficiently with the long fragment in the more highly degraded eDNA sample. The authors find well over 2000 species from 12-15 animal phyla in all of these samples, which seems rather phenomenal, but probably in line with other studies of metabarcoding, which now use a standard approach based on the Usearch pipeline for these purposes.

3. >>> The bioinformatics processing and OTU clustering approaches employed here will provide variable numbers of OTUs according to clustering ⁴ and abundance filtering ⁵. By adhering to stricter abundance filtering as highlighted below, we now present lower levels of richness in alignment with empirical data of Chironomidae diversity (historical data from Llyn Padarn 2003-2013) and low-levels of contamination according to the positive controls (please also note that the original 2000 number referred to a composite aggregation of both eDNA and community DNA and not unique OTUs, hence seemingly inflating the number of OTUs found). Nevertheless, the unknown entity of simultaneous sequencing of protist diversity would also be predicted to boost realised levels of diversity during an annual survey of lake ecosystem biodiversity.

The paper seems rather difficult to read because of the poor structure, even if reading the Methods first (which are appended to the end).

4. >>> We apologise if the format of appending the method section to the end of the manuscript lacks clarity, but it is the required journal format. Regarding the structure, the Reviewers Two and Three commended the quality of the writing. Nevertheless, throughout the revision process, we have refocused again on enhancing clarity through structural changes and hope that these now work in getting the message across for all the reviewers.

I was struggling in particular about how the distinction is made between the 'eDNA' and 'community' samples. The first is by filtration of the water, the second by scooping off the exuviae of chironomids from the surface, if I understand correctly.

5. >>> The reviewer is indeed correct in their understanding and we have now made this explicit at first mention. The corresponding author has also published a recent review ⁶ that contributes to this discussion for further clarification if required.

In both cases small individuals including microbes are filtered out of the water. Hence the DNA is not 'free' in most cases. This is perhaps not terribly important, if the aim is to gather the species diversity of a lake from a few litres of water, but it's not 'eDNA' in the strict sense.

6. >>> Here, we refer to nomenclature defined in ^{3,7,8} as "environmental DNA is extracted directly from the environment, without prior isolation of the organism itself" and "eDNA can originate from a variety of sources, including but not limited to, feces, saliva, body mucus, shed skin cells, urine etc." The nature of macrobial eDNA itself is considered to include both extracellular (free) molecules as well as cellular forms, with the cellular form also transforming to the extracellular due to degradation ⁹. We also predict that the diameter of filter membrane used for the experiment (0.45µm) will retain a diversity of macrobial eDNA sources whilst avoiding excessive clogging of filters (see below). Furthermore, the nature of the DNA extraction protocols, including a digestion step (Proteinase K) was used to ensure that cellular eDNA was extracted in addition to extracellular sources. We also endeavour to make this clearer for the readership on Page 3, Lines 52-55 and throughout the Methods section.

The authors made no attempt to characterise their samples, not even for the most basic parameters such as DNA concentration and average fragment length.

7. >>> The average fragment length (of the protein coding gene) is stated to in the methods section (658bp and 235bp) and we now provide tables in the supplementary material listing the concentration of extracted DNA for all samples (Supplementary Tables ST4 - ST5).

The degree of degradation of the two samples is characterised by comparison of the short and long PCR, but this test is compromised by the selection of different primer sequences in one of the two primers, to include a more taxon-specific sequence.

8. >>> According to the concerns presented by the reviewers, we now only report on the qualitative data derived by the long fragment and focus our comparative analyses for the Chironomidae entirely on the short fragment. In order to provide a more balanced perspective, we have also discussed the issue of primer bias, in addition to the lack of any obvious bias in amplification across the Chironomidae tree of life (Supplementary Figure SF8).

The analysis is then focused extensively on the species richness and turnover with the seasons, resulting in the intuitive result that there are slightly more species in the warmer months. I don't think this is a big deal and should be toned down.

9. >>> In the revised discussion, the commentary on species richness of the Chironomidae, measured via the analysis of the exuvial community DNA has been toned down accordingly, although we do still state the novelty of the analyses. Moreover, the discussion of the temporal turnover was mainly aimed at verifying our findings which were in accordance with established knowledge of temporal turnover of freshwater ecosystems. We have maintained the insights derived from the beta diversity analysis of the eDNA, since this was one of the principal aims of the study. We have also added a theoretical framework for the dynamics between communities, environmental DNA and beta diversity throughout the seasons, representing a fuller integration between the different forms of DNA in the discussion, Page 17, lines 373-382.

The more important question is about the identity of the various chironomid species, and how they individually respond, in particular in relation to their close relatives (on the tree built from these data). There is the potential for lots of interesting analyses, but we are left with a very simple tree that mainly illustrates the presence/absence of particular species in the two samples.

10. >>> We would like to highlight that the purpose of the tree was to provide a phylogenetic representation of the taxonomic coverage resulting from the use of the different primer sets, to illustrate a lack of primer bias across the chironomid tree of life. Nevertheless we fully agree with the reviewer that the dataset provides potential for more analyses and have performed these at the individual OTU/species level, in relation to DNA source, methodology and ecological relationships within the current datasets and across UK Chironomidae biodiversity and provide full details below.

It is not linked to known species, although it would be very good to establish if the species detected here are indeed those known from the lake.

11. >>> In the original version of the manuscript, we provided companion analyses of the community derived DNA and also environmental DNA to provide an explicit molecular genetic comparison to address this point. Nevertheless, now we have engaged with taxonomic experts in order to identify up to 800 individual chironomid exuviae that were partitioned from the April, May, July and September sampling points, in order to gain at least genus level identifications for the physical exuvial samples (Methods text Page 25, Line 560; Taxonomic identification of invertebrate community samples). Given the extremely challenging nature of chironomid exuvial

identification, often only genus level taxonomy can be attained. Nevertheless, these new analyses have facilitated a direct comparison between the biodiversity estimates derived from the community DNA, environmental DNA and taxonomic approaches. These are now presented in Figure 1 and discussed throughout the text. Indeed, the new analyses do show clearly that the species detected using molecular approaches overlap substantially (61-65%) with traditional taxonomy approaches, i.e those known from lake.

This leads to the wider question about the identity of the ~2000 species found, whose identification greatly depends on existing public databases. What is the confidence? The reader wants to be convinced that this massive sequence variation is a meaningful description of the life in the pond.

12. >>> Following the abundance filtering approaches suggested by Reviewer Three, the alpha diversity of the samples within the study have all been substantially reduced, although the principal beta diversity analyses are representative of the unfiltered analysis. For the remaining OTU's, we employed a combination of carefully curated blast approaches for broad taxonomic assignment, complemented by individual blast scrutiny (Alignment against barcode sequences obtained herein and from public records), phylogenetic assessments and BOLD species assignment (>99% hits) to achieve robust taxonomic identifiers for our dependent variables. Please refer for example to Page 25, Lines 548-555 for clarification in the manuscript.

My suggestion is to slim down this paper mainly by removing the somewhat repetitive and unconvincing analysis of temporal turnover and to focus more on the methodological aspects of community characterisation.

13. >>> As requested by the reviewer, we have streamlined commentary on the temporal turnover of the Chironomidae communities. Moreover, we have focused further on the relationships between sequence abundances, methodology and time at the individual OTU level using a generalised additive model (GAM). We have also investigated associations between species/OTU abundances within the study system and chironomid species across the United Kingdom. The analyses show that different species yield different levels of eDNA, eDNA abundance differs throughout the year, the relationship between eDNA and community DNA is heterogeneous across different OTU's and OTU read frequencies within the study system are significantly positively associated with Chironomidae species frequencies resulting from meta-analyses of 435 lake ecosystems throughout the UK. The new analyses are integrated into the methods, results and discussion fully.

How much of the eDNA is derived from living things, and is it really possible to do barcoding on free DNA?

14. >>> In our study, we executed an experimental design that sampled community DNA (exuviae) derived from recently living/emergent species (within 48 hours of emergence) and simultaneously sampled aqueous eDNA, using the same molecular markers across samples to enable this comparison. There is also a substantial volume of literature already in existence

focusing on the analysis of, and reviews concerning microbial eDNA from mesocosms⁹ and different natural and semi-natural ecosystems^{1,2}. Collectively, the state-of-the-art of the field clearly highlights that it is possible to perform (meta)barcoding on eDNA^{5,10,11}, but what is lacking from the existing literature is an insight into temporal relationships between the living community and eDNA in the wild, as we endeavour to provide here.

How much 'false positives' are obtained by the Usearch process? How secure is the identification, and do these two different primer sets arrive at the same identification?

15. >>> Regarding false positives and the USearch process, please refer to answer 12. and further clarifications for Reviewer Three. Regarding the different primer sets and identification - by referring to the phylogenetic hypothesis (Previously Supplementary Figure S6, now S8) derived from a quality controlled and manually created alignment, we provide phylogenetic evidence of fully supported, monophyletic clades defining sister OTU's delimited by the different primer sets. Therefore, the two different primer sets result in the same identification, afforded in most cases by identical phylogenetic placement across the chironomid tree of life.

In addition, rather than drawing the NMDS plots that are very popular in metabarcoding studies but ultimately rather uninformative, perhaps a focus on particular species and their turnover could be more interesting. I don't think the power of metabarcoding is primarily in the quick-and-dirty plot of species diversity (even if that was ok for the early papers in the field demonstrating the feasibility of metabarcoding in principle). This would fall short of conventional studies in community ecology that rely on proper identification and evolutionary analysis of the components.

16. >>> We share the reviewer's enthusiasm for appropriate analysis for meta-barcoding data in the context of community ecology and are grateful for their constructive suggestions. We would also like to highlight that NMDS plots, in our opinion, are a useful way of visualising species turnover, but also provide the visual backdrop to test the statistical significance of relevant ecological groupings (e.g. ANOSIM variance analyses as in the study). In addition, we assessed individual species/OTUs in relation to methodology and time using a GAM, as mentioned in answer 13. NMDS, as the reviewer indicates has been used previously to provide biodiversity insights from high throughput sequencing data, allowing us to draw more accurate comparisons with other's work, especially as our filtering of water samples and resident eDNA, combined with community analysis is quite novel and benefits from standardized analytical approaches. In our revised version, we aim to provide a balance of established ecological biodiversity descriptors (according to the focal aims of the manuscript) as well as novel OTU/species level investigations in relation to methodology, time and linkages to chironomid biodiversity at the local, including across a broader landscape scale (Answer 13). By combining such approaches, we aim to provide deeper insights into the relationship between eDNA and community DNA at the species/OTU level, identify important temporal trends in eDNA community diversity at alpha and beta levels and complement with a more balanced discussion discussing the temporal dynamics of eDNA and the living community across the annual scale.

Therefore, we would like to reiterate our thanks to Reviewer One for providing the relevant insights into how we could integrate additional analyses into our existing findings.

Reviewer #2 (Remarks to the Author):

A. The authors applied eDNA metabarcoding from lake water samples to screen for changes in diversity over the course of one year. Focus was on chironomid diversity as well as of the lentic community. Furthermore, the authors used two different COI primer combinations (658 bp vs. 250 bp) to test for their efficiency in metabarcoding. The data reveal an enormous diversity of OTUs / taxon richness and using the data the author show clear annual changes in biodiversity patterns and thus the power of eDNA as a tool for biodiversity assessment. I enjoyed reading the manuscript but have a few major points to raise concerning data analysis and interpretation (see C-F).

B. High! Great interest to broad readership!

17. >>> Many thanks!

C. Sampling, laboratory and bioinformatic methods mostly sound but still with few major issues and at least missing important information at several parts.

- Was it always the same person performing sampling, extraction and PCR?

18. >>> Yes, IB performed all components of the field and lab work as is also stated in the author contributions section (Page 31, Lines 784-785)

Please elaborate further on standards (The room where water was filtered was PCR-free, but what kind of other samples are handled in this room? Was the room prepared for eDNA extractions, e.g. by filtering air and UV treatment? Same for the building where extractions were performed; where protective clothes used).

19. >>>>> Prior to filtration, the room was not used for handling samples. Both the filtration and the extraction room were cleaned and thoroughly bleached before use. For extractions, a designated fume hood was used (in the previously mentioned clean room). A separate room (in a third building, not previously used for handling DNA of any sort, bleached as described previously) was used for library preparation. Fresh clothing and clean-room only lab attire was used during extractions and library prep. Furthermore, the negative controls (equipment, membrane filters and reagent negative controls) did not show any band after amplification. After high throughput sequencing only two OTUs were found in each amplicon, which were bacteria (COIS amplicon) or low numbers of reads (gastropoda <240 reads, diptera < 10 reads, across blank samples). The above verify the good standing of collection and decontamination protocols used.

Since you used reusable filter funnel units, how did you make sure that they were not contaminated when used for the second time?

20. >>>>> In the Methods, Page 20, Lines 443-444 refer to supplementary information (Supplementary Methods SI.3) describing fully the decontamination protocols. As explained in detail in SI. 3, Pages 2-3, Lines 43-52, all laboratory equipment (including filtration units) would undergo multiple rounds of decontamination. Additionally, (Page 3, Lines 53-61) blank equipment (distilled water) were collected, which were PCR amplified and sequenced to test for possible contamination.

- Please add information on how you handled clogging filters (we experience this issue). How many filters were used per sample?

21. >>> During method testing we had experienced clogged filters, and we were prepared to use additional filters if needed. Nevertheless, during filtration of samples used in the study, we didn't have that problem. This could be because the samples were carefully collected from the near surface of the oligotrophic water body, additionally avoiding any sediments.

"For large samples, ~1g of dry invertebrate material was subsampled from the bulk, whereas for some low-density winter samples, the entire sample (<1g)".

By using different amounts of tissue, you possibly introduced bias due to oversampling some and undersampling other samples. Why did you not use the same amount of tissue (<1g) for all samples?

22. >>> The simple answer to this question is that for some of the low-density winter samples, there was not 1g of exuviae available and so the whole sample was used. For the remaining high diversity/abundance samples, subsampling of 1g of community material emulated the subsamples that are routinely taken for the CPET taxonomy-based approach¹² that is used to assess, and report on chironomid diversity in relation to biomonitoring/ecosystem status to the European Union. Therefore, the subsampling represented a pragmatic approach providing a balance between effective and proven ecological sampling, whilst also not overloading DNA extractions with excess material and potential inhibitors from invertebrate cuticles.

- Also: Please state that a sampling permission was obtained from Snowdonia National Park officials.

23. >>> We now state in the text, Page 19, Lines 427-428 "As only water and exuviae (shed skins) were collected and the work was performed in collaboration with the EA and NRW (that currently manage the site), a permit was not required."

- Each sample was amplified in triplicate. Please provide some more details on library preparation and how individual samples were tagged (issues of amplification bias due to indexes possible?

24. >>> Please refer to Answer 46 below, in order to see full details of library preparation and in particular, the use of the same Illumina tailed degenerate primers across all samples (thereby removing the potential for index related amplification bias), prior to individual identification via end tagging.

- It is very important to have a table that lists read numbers per replicate. Please include these information!

25. >>> As requested, the data are now presented in Supplementary Table 1.

- Please explain why only singletons were removed and not reads up to a certain (low) threshold in order to avoid biodiversity overestimation due to reads that are PCR artifacts.

26. >>> For full coverage of additional bioinformatics abundance filtering, please refer to Answer 45. in response to Reviewer Three's questions, thank you.

- While all tools used represent absolutely high-end tools for bioinformatic analysis, there are some issues that should be changed. Clustering was performed independently for the two read data sets. However, for a much better comparison the authors should do clustering together for both data sets (COIF, COIS).

27. >>> This is an interesting suggestion, but in our experience and that of others in the microbiome field ¹³, introducing heterogeneity (e.g. length) into OTU clustering can sometimes create chaotic results and so for our primary analyses, we elected to cluster the loci independently. Nevertheless, by referring to Answer 15. above, combining the focal Chironomidae OTUs by sequence alignment and phylogenetic reconstruction, we gain a clear phylogenetic perspective of the OTU's delimited by the different loci, derived from independent, clean OTU calling.

- When comparing efficiency of the long Folmer fragment (watch out, mistakenly it is written HCO2189 instead of HCO2198 at one point) to the short fragment the improved efficiency cannot solely be assigned to the length. The primer used (from the Carew et al. study) has a degenerate base ("H") at the very last bases (not present in the Folmer primers). Thus, improved efficiency can be a result of this! I would definitely expect more OTUs through this approach.

28. >>> We have corrected the HCO2198 issue and also now discuss more clearly the issue of potential primer bias. Moreover, on the basis of the cumulative comments from the reviewers, we have further scrutinised the disparate levels of coverage across total biodiversity and the focal chironomid group across COIF and COIS and now focus our comparative community and eDNA Chironomidae analyses, only focusing on COIS. We do present the data derived from COIF more broadly and in the supplementary information, but also draw attention to the overt lack of coverage of the long fragment from the sequencing runs, which also points to low levels of available target in otherwise, well covered samples.

- Please use a full stop and not a comma as a decimal separator in Fig. 1.

29. >>> Decimal separators have been harmonised to full stops throughout.

D. Mostly correct.

But:

- While it is useful apply rarefaction, limiting analysis due to the lowest read number (476 reads) is a concern. I would advise to analyze the data not including the sample with such a low read number to test if results differ. 476 reads (COIF) is by far too low for an NGS OTU analysis.

30. >>> During the reanalysis of the data set and the removal of the comparative analysis of community and environmental DNA derived from COIF, the minimum level of rarefaction now stands at 4000, in order to address the reviewer's concerns.

- I suggest using not the Phred Quality Scores but the USEARCH scores for quality trimming.

31. >>> USEARCH applies quality filtering used a maximum expected error (maxee), which is actually calculated in relation to Illumina Phred scores (for full background, please see USEARCH manual http://www.drive5.com/usearch/manual/exp_errs.html). A maximum error of 1 was used for quality filtering in both the first submission version and in the new. We apologise for not referring to this specifically in the first version of the manuscript. Accordingly, we have now added in the methods section Bioinformatics and statistical analysis Page 23, lines 513-515. "Using USEARCH (fastq_maxee = 1) sequences with a maximum expected error (maxee) > 1 were discarded. Maxee is the expected number of errors as sum of the error probabilities (provided by Phred scores)."

- Page 12, line 7: "Both amplicons presented the same annual pattern, with a decrease in species richness over winter and a summer increase related to rising water temperature (Fig. 3a-b, orange lines)." This is not correct. The orange line clearly does not support a clear pattern of increase during the summer months and decrease during winter months: for COIF, community OTU richness is high in January, even higher than in some summer months. For COIS, there are just two data points for community OTU richness between September and May, which does not allow inferring clear patterns from the data. Thus, I suggest to refrain from drawing best fit curves when the data do not permit it and rephrase the sentence accordingly. For eDNA (blue line and dots), there is a gap of data points between February and June in the COIF dataset. Again, it is not advisable to draw a best fit line. Also, the fact that these data points are missing is not discussed in the manuscript (they are present in Fig S3). The emergence pattern of chironomids can hardly explain the complete absence of any DNA in the water and rarefaction analyses should also not cause the observed pattern,

especially since lots of OTU were present in February. Please discuss possible reasons for the lack of data.

32. >>> This aspect of the manuscript has changed substantially as a result of the data reanalysis and so now the contentious sentence has been removed from the text. Figure 3 does feature best fitted lines, but we now also complement with an explanation why a number of winter sampling points did not yield genus richness data. The reason for the lack of data across the winter period was simply because they were very very small numbers of exuviae collected across this time period and subsequently, they were not sequenced adequately in a mixed amplicon pool. Conversely, during the summer months the number of exuviae (representing substantial biodiversity) would fill a number of 50ml Falcon tubes. We have also now made this clear in the results, discussion and where appropriate on the figure legends to reassure the reader that these are not necessary missing data points, but instead times when richness was very low.

E. Part of the conclusions are correct but mainly available data should be included (tables on libraries / read numbers / gram of tissue for bulk samples), clustering be performed together for consistency (COIS and COIF), possibly use a lower threshold to avoid including OTUs that are PCR artifacts, avoid including the n=476 samples for rarefaction and reanalyze. Explain missing values in the time series and more carefully interpret the fitting lines of the model due to the missing data points (see comments above). No new experiments needed, no other software approaches needed.

33. >>> Data on libraries and read numbers are now presented in supplementary table ST1 and gram of tissue for bulk samples are provided in the main text. Please also see comments 27 for co-clustering of the data and for full coverage of additional bioinformatics abundance filtering, please refer to Answer 45. in response to Reviewer Three's questions. Comments 32. also cover the n=476 issue and the apparent missing data points in the model data.

G: Appropriate

H: Abstract, Introduction and Conclusion extremely well written and clear!

34. >>> Many thanks!

Minor points:

Introduction:

"eDNA ecology" is not the right term, please consider rephrasing the sentence. E.g. "...several factors of eDNA research ..."

35. >>>> The sentence has been rewritten accordingly.

Discussion:

Page 10, line 15: " In contrast to previous analyses of macroinvertebrate species from eDNA samples 19,20, we employed a metabarcoding approach instead of PCR-based detection". This is not entirely correct, since metabarcoding is still a PCR-based approach. Please remove this sentence or rephrase it accordingly.

36. >>> Here we were not referring to PCR-free vs. PCR based work. Instead, we were referring to work employing quantitative PCR (qPCR) as the final means for species detection compared to metabarcoding work, where (even though PCR amplified) the researchers rely on HTS results to confirm positive detection. Moreover, qPCR work can only focus on a single or a low number of species while metabarcoding allows detection of a wide range of diversity. We have adjusted the text accordingly as requested by the reviewer in order to clarify this, Page 12, Lines 248-251.

Page 12, line 1: " Other possible approaches for HTS of eDNA are direct shotgun sequencing of DNA extracts or use of capture probes". While the mentioned techniques are very promising without doubt, please keep in mind that the amount of DNA needed for direct HTS sequencing is relatively high and cannot be easily obtained by sampling few ml or l of water. Please rephrase this sentence accordingly.

37. >>> This sentence now reads:

"Other suggested strategies for enhancing HTS of eDNA (where concentrations are sufficiently high) involve direct shotgun sequencing or use of capture probes^{26,40}." Page 15, Lines 326-328.

Page 8: Dissolved Oxygen "D.O "

38. >>> The appropriate abbreviations, with correct decimal places have been instated throughout the text.

In general: When using the scientific name "Chironomidae" there is a capital C. If you write about chironomid samples, it is a small c.

39. >>> These have been corrected in throughout where appropriate, thank you.

Reviewer #3 (Remarks to the Author):

Bista et al. "Annual time-series analysis of aqueous eDNA reveals ecologically relevant dynamics of lake ecosystem biodiversity".

The authors investigate the use of eDNA metabarcoding of lake communities in parallel with bulk sample metabarcoding of Chironomids. They do temporal analyses of 16x2 samples taken over

1 year. Although the approach of temporal assessment of diversity by eDNA is fairly new, Biggs et al. (2015). *Biol. Cons.* analysed newt eDNA collected 4 times during the breeding season.

40. >>> We are of course well aware of the excellent work performed by Jeremy and colleagues and the corresponding author frequently highlights this paper on the conference circuit, given its importance from a regulatory perspective in the field. Nevertheless, and for the benefit of the editors, Biggs et al. ¹⁴ employed the use of qPCR to look for a resident single species over a 3 month period, in order to enhance detection capability. Here however, we have used metabarcoding to investigate community diversity of numerous taxa that follow phenological patterns of life, death and decomposition over an annual cycle. Therefore, to our knowledge, the novelty of the work is clear.

The results of Bista et al. are interesting and the paper is generally very well written, but there are some major considerations to be dealt with.

>> Thank you and see below.

MAJOR COMMENTS

1. The consideration and careful trimming for PCR and sequencing errors
2. PCR tagging system in two rounds of PCR
3. Choice of target region (COI)

41. >>> We will deal with each of the three points sequentially below, thank you.

1. The consideration and careful trimming for PCR and sequencing errors.

It is well known that numerous errors occur during PCR and sequencing (Schnell et al. 2015. *Mol. Ecol. Res.*, Coissac et al. 2012. *Mol. Ecol.*).

The authors do not describe adequately how errors were dealt with besides that chimeras were removed. The fact that they retrieve "an impressive amount of diversity from eDNA samples" and "with up to 2730 OTUs from one amplicon" should cause some critical consideration. I do not think it is unlikely that high diversity of invertebrates could be obtained by eDNA, but one must be cautious. Especially when the numbers is this high in a northern European lake in one liter of water!.

42. >>> For simple clarification here, we must also acknowledge the sampling represented 2x1litre water samples, but importantly, from throughout the year (including all diversity and not restricted to invertebrates), thereby capturing the annual biodiversity of the lake ecosystem.

Fx. the authors retrieve taxa in high numbers that are very rare in lakes (Arachnida!, besides the few species such as *Dolomedes* and *Argyronecta*). One approach is to check if these are variants of the very abundant insect reads.

43. >>> The corresponding author has checked the identification of the terrestrial taxa noted in the manuscript and we are convinced that their presence in the eDNA samples are valid. For example, *Segestria* (likely *senoculata*) never sequenced in the lab before, was present in 12/16 samples and represented by 3753 reads. *Segestria senoculata* occupies shale/slate waste and is found commonly around the study site. Similar also for *Xysticus* (likely *cristatus*; 8/12 samples, 1858 reads) is a very common Thomisidae species found in N Wales. In order to balance this discussion, we cover the occurrence of terrestrial taxa, similarly checked carefully now in the results and discussion. We are also aware that terrestrial taxa feature in another eDNA Nature Communications manuscript under final review that investigates eDNA in river ecosystems and so this particular finding has been corroborated by an independent lab (Deiner, Altermatt and colleagues).

It would be very obvious to provide a full species list (especially for the COIF region that can go to species level) to see if/how many "exotic" species turn up. This can be used to evaluate the amount of recovered MOTUs compared to known species diversity in the mock sample. Is significantly more/less MOTUs/unique sequences being recovered than there are "species" added in the positive control sample ?

44. >>> Given the lack of coverage of COIF, we have now removed our targeted analysis of this fragment from the main body of the manuscript. Nevertheless, referring to Supplementary Table ST3, we can see that between 0.007-0.026% of reads were non-target reads in the positive control sample and the COIS negative control returned no false positives. In combination with our re-analysis of the full dataset (see below, inspired by Reviewer Three), we are now confident that the presented richness accurately depicts an appropriate level of OTU diversity for a temperate lake ecosystem.

Another obvious approach is to look into the non-target reads of the positive controls and use this as trimming. Some similar approaches are in Port et al. (2016). Mol. Ecol. I do realize that the authors find separation of season in the NMDS analyses and that such an overall ecological signal could be retained regardless of stringent trimming, but errors still needs to be taken into account, and how this influences the results.

45. >>> Indeed, we very much enjoyed the contribution by Port et al. ⁵ and appreciate that the sensitive nature of high throughput sequencing may yield false positives and low-level variants in your datasets. Therefore, in order to address this concern, we investigated a dual approach as to how best to trim the data in an appropriate fashion that would reflect both empirical evidence derived from the positive and negative control samples, but also an accurate representation of genuine biological diversity in the lake ecosystem. Therefore, we investigated OTU richness patterns by comparing the raw data analysis, with trimming reads that were present in less than 0.01% and 0.02% of each sample and then compared these results with levels of actual richness that were detected from the lake according to traditional taxonomic approaches. Following the assessment, we thereby observed that a trimming strategy of 0.02% returned values of richness that were concordant with the expected taxonomic richness of the lake, but also provided a trimming window that was in alignment with the empirical observation

of false positives observed in the control samples (0.007-0.026%). We would therefore like to thank the reviewer for the suggestion and present in the revised paper, a completely reanalysed dataset in accordance with this additional level of quality control, based on both taxonomy and empirical evidence. Accordingly, levels of alpha diversity have fallen. We also agree with the reviewer that our findings reporting the dynamics of beta diversity remain unchanged, and these are now augmented by the additional species level analyses inspired by Reviewer One.

2. PCR tagging system in two rounds of PCR.

As I understand, the authors use a first round of PCR with the Illumina adaptor and a second round with individual indexes? This approach could cause problems since the individual PCRs (samples and replicates) are not tagged from the first round of PCR. This can severely increase cross-contamination risk if done in parallel for this many samples, and if so, there is no way of checking from which sample a sequence originally came from.

See also DeBarba et al. 2014. Mol. Ecol. Res.

46. >>>> Over the past 10 years, we have use combinations of different tagging strategies, including the use of initially labelled oligonucleotides in either single, or dual step PCR. Nevertheless, as the field has matured, there is intuitive and also now abundant empirical evidence to show that if you perform initial PCR reactions with different combinations of tails to identify individual samples, the heterogeneity in the oligonucleotide is reflected in variation in amplification across samples^{15,16}. Therefore, and in alignment with the maturation of the field of microbiome analysis, we prefer here to adhere to robust laboratory protocols and the amplification of different samples with uniform taxon specific primers, followed by a modest number of PCR cycles to adapt Illumina universal tails and unique identifiers onto each individual sample. If, as the reviewer states, that we would have no way of knowing which sample was which, these data and that of hundreds of other studies that are currently adapting an identical strategy would yield ecological meaningless data. As the reviewer has already highlighted, our analyses represent tractable ecological dynamics and we would also like to respectfully highlight the following points to make our response to this constructive, but serious, criticism absolutely clear to the reviewer and Editor.

a. In our library preparations, we use IDT True Grade primers, recommended as the gold standard in oligonucleotide accuracy by a number of UK Genome Centres:

<https://www.idtdna.com/pages/decoded/decoded-articles/competitive-edge/decoded/2012/09/21/reduced-barcode-contamination-using-oligonucleotides-with-trugrade-processing> - Therefore, primer synthesis errors are minimised.

b. Our labelling strategy (Nextera octamers - optimised differences between tags) and that presented in Miya et al., is the recommended Illumina protocol for amplicon bar-coded library constructs. Moreover, our forward primers feature the 5xN feature between universal tail and template specific primer, which allows optimal cluster detection on the MiSeq platform.

c. Our adapted amplicon libraries proceed direct to Illumina sequencing, negating any blunt end ligation steps and potential tag jumps.

d. Both our positive and negative controls showed no evidence that samples were somehow mixed along the workflows that we have employed.

Therefore, we have no reason to believe that our approach could cause problems in relation to our data analysis and downstream ecological synthesis.

3. Choice of target region (COI).

The use of COI for DNA metabarcoding studies have been criticized (Deagle et al. 2014. Ecol. Letters). The primers are not very universal when it comes to equal amplification chance of all taxa, which can skew the final results on community composition and thus the interpretations of these results. The authors need to address this in the discussion and do some in silico testing of which taxa will be over/under represented in PCRs and thus in the final sequence reads.

>>>> 47. We are indeed aware of the limitations of the COI marker for meta-barcoding studies and of the (Biology) Letters paper. Nevertheless, for this particular body of research we required (a.) metabarcoding primers that would amplify across a broad range of taxa (in particular, lake occurring taxa), (b.) a marker enabling the best annotation power for macroinvertebrates and in particular, the Chironomidae, (c.) a combination of two primer pairs providing different length amplicons. On this occasion, the maturation of the COI database in association with generally broad amplification across taxa and especially arthropods and the flexibility to design primers to investigate different lengths made the choice of COI for this study clear. Moreover, the use of the COI locus facilitated the use of the Chironomidae specific combinations of primers that amplify the focal COIS amplicon in the study. If any other marker had been used, we would not have been able to provide taxonomically comparable data to compare with taxonomy and existing approaches. In order to make this clear, we provide our rationale now in the Method section, Page 21, Lines 473-477 and also highlight reference to the limitations of the Folmer primers for meta-barcoding between Page 14, Lines 302-303 in the Discussion. Rather than in silico analyses, we now also present the taxonomic overlap between diversity recovered by traditional taxonomy and the COIS marker in Figure 1, providing an empirical comparison of comparability of the locus and biodiversity.

MINOR COMMENTS

#Abstract:

The first sentence is not very clear. The "sensitivity" of what? Also "free" is not a good word as you don't know if the eDNA is in fact extracellular.

48. >>> We have changed the first sentence of the abstract to read:

"The use of environmental DNA (eDNA) in biodiversity assessments offers a step-change in sensitivity, throughput and simultaneous measures of ecosystem diversity and function."

What is meant by "efficacy of the ecological signal"?

49. >>> This sentence no longer exists in the revised manuscript.

"for elucidating drivers of biodiversity change over time" should be changed to "for tracking biodiversity change over time". Think this is more correct

50. >>> The final sentence and the abstract has been reworded accordingly, many thanks.

"enhancing efficiency"? of what?

51. >>> "detection efficiency" of biodiversity and this is now made clear in the final sentence of the abstract.

#Introduction:

"it is essential that we employ new, reliable and cost effective methods for biodiversity assessment". Why is "new" relevant here?.

52. >>> This sentence has been rewritten to include the text "there is a need for increasingly reliable and cost effective methods for biodiversity assessment" on Page 3, Lines 43-44.

"shed skin cells, urine, faeces or saliva of taxa". Taxa should be changed to organisms.

53. >>> The sentence was a focus of a previous reviewer's comment and so the entire sentence has been adjusted thus:

"Sources of eDNA include sloughed skin cells, urine, faeces, saliva or other bodily secretions, and consist of both free molecules (extracellular DNA) and free cells."

#Results:

Define COIF and COIS the first time (Full and Short)

54. >>> Definitions for COIF and COIS are now made at first mention in both the methods and results section, to improve clarity for the reader.

"mean number of reads {plus minus}SD: COIF: 269,769 {plus minus}57,427; COIS: 259,723 {plus minus} 85,437". Mean number pr sample? Please specify!

55. >>> Per sample has been clarified on Page 6, Lines 126-127.

"393(COIF) and 547 (COIS)". Write the full number (393.000 etc.) to be consistent with above text.

56. >>> The full numbers now feature on Page 7, Line 133.

"cubic model explained the highest proportion of diversity in our data". Don't you mean that it explained the variation in our data?

57. >>> Yes, but this analysis no longer features in the manuscript.

"showing ecologically meaningful sub-groupings within "winter" and "summer" samples". Please explain and expand on how this was meaningful.

58. >>> This paragraph has been rewritten on Page 9, between lines 186-197, with the phrase removed and the results of the beta diversity analysis presented in an objective fashion.

"OTU richness for both amplicons ($p < 0.05$), and between D.O. and community DNA". What is D.O??

59. >>> D.O. means dissolved oxygen and the abbreviation is now clarified in both the methods and results section upon first mention.

"throughout the Chironomid tree of life". Would use another word, fx. across the Chironomidae

60. >>> The use of Chironomidae as a descriptor has been substituted throughout the manuscript when "tree of life" was used previously.

#Discussion:

"Our findings yield a clear characterisation of temperate lake ecosystem-wide biodiversity". Please specify why this is so?

61. >>> this sentence has been rewritten as:

"Our findings yield an informative characterisation of temperate lake ecosystem-wide biodiversity" on Page 12, between Lines 255-256.

"with likely co-variance with productivity". Specify productivity! Is it primary productivity?

62. >>> Following our revisions, this sentence no longer features in the manuscript.

"Evidence supports the preferred use of longer amplicons to maximise clear and temporally short-term shifts in community diversity". I think you mean that your results (in stead of "evidence") support the preferred use of.....

63. >>> Following our revisions, this sentence no longer features in the manuscript.

"Variations in community composition (β -diversity) that are observed over April and November could either reflect seasonal turnover or possibly be attributed to lake inversion effects". I guess lake inversions would in fact be characterized as one of the drivers of seasonal turnover.

64. >>> We agree and have reworded this sentence as:

"Changes in observed community composition (β -diversity) over April and November (Fig. 2, Supplementary Fig. 6) most likely reflects seasonal turnover, possibly attributed to lake inversion effects." on Page 16, Lines 347-349.

#Methods:

"it was demonstrated that the latter protocol yielded the highest number of DNA copies". For what species/taxa??

65. >>> this sentence has been reworded to make clear that Renshaw et al. were investigating specific eDNA targets as:

"In Renshaw et al. 52 it was demonstrated that the latter protocol yielded the highest number of DNA copies of targeted eDNA fragments."

"Such a strategy ensured that the analyses featured the maximum number of independent samples, covered with an appropriate depth of high quality sequences according to their predicted biodiversity." I cannot see how this is obtained by rarefaction/normalization?

66. >>> This part of the manuscript has been rewritten thus:

"For downstream analyses, the appropriate depth of coverage per sample was determined according to rarefaction analyses - OTU accumulation vs. sequence coverage curves generated in QIIME. Samples were subsequently normalised using rarefaction in QIIME at appropriate depth for each amplicon 68." On Page 25, Lines 557-559.

#Figures:

Figure 1. The colours are confusing. Use the same colours for the same designations.

67. >>> This Figure no longer features in the manuscript.

Figure 4a. Only one sample from community DNA in winter ??

68. >>> Figure 4a does not feature in the revised manuscript and limited representation of certain samples in the remaining manuscript are discussed in relation to low sequence coverage, associated with very low abundance samples (c.f. answer 32).

#Supplementary Material:

The "CPET technique" is not defined

69. >>> Please see the start of the Supplementary Information SI.1:

“SI.1 Emergence patterns of Chironomidae and the Chironomid Pupal Exuviae Technique (CPET) technique.”

"10% Trigene" is not defined

70. >>> Trigene is defined as “(Ammonium chloride & hydrochloride, Medichem Int.)” in the Supplementary Information, Page 3, Lines 45-46.

"Possibility for collection of larger and more ecologically relevant water sample (2L)" This lacks support in the literature I think.

71. >>> This sentence has now been removed from the manuscript.

"Optimal pore size for collection of smaller DNA molecules and microorganisms" But you are not targeting microbes.

72. >>> This sentence has now been removed from the manuscript.

Figure S1: What is "low numbers of OTUs in the dataset" ??

73. >>> This figure and legend no longer feature in the manuscript.

Figure S2: Needs scale

74. >>>> The figure has been adjusted to include the scale

References

- 1 Bohmann, K. *et al.* Environmental DNA for wildlife biology and biodiversity monitoring. *Trends in Ecology & Evolution* **29**, 358-367, doi:10.1016/j.tree.2014.04.003 (2014).
- 2 Thomsen, P. F. & Willerslev, E. Environmental DNA - An emerging tool in conservation for monitoring past and present biodiversity. *Biological Conservation* **183**, 4-18, doi:10.1016/j.biocon.2014.11.019 (2015).
- 3 Barnes, M. A. & Turner, C. R. The ecology of environmental DNA and implications for conservation genetics. *Conservation Genetics* **17**, 1-17, doi:10.1007/s10592-015-0775-4 (2016).
- 4 Fonseca, V. G. *et al.* Second-generation environmental sequencing unmask marine metazoan biodiversity. *Nature Communications*, DOI:http://dx.doi.org/10.038/ncomms1095 (2010).
- 5 Port, J. A. *et al.* Assessing vertebrate biodiversity in a kelp forest ecosystem using environmental DNA. *Molecular Ecology* **25**, 527-541, doi:10.1111/mec.13481 (2016).
- 6 Creer, S. *et al.* The ecologist's field guide to sequence-based identification of biodiversity. *Methods in Ecology and Evolution*, n/a-n/a, doi:10.1111/2041-210X.12574 (2016).

- 7 Dejean, T. *et al.* Persistence of Environmental DNA in Freshwater Ecosystems. *PLoS ONE* **6**, e23398, doi:10.1371/journal.pone.0023398 (2011).
- 8 Rees, H. C., Maddison, B. C., Middleditch, D. J., Patmore, J. R. M. & Gough, K. C. The detection of aquatic animal species using environmental DNA - a review of eDNA as a survey tool in ecology. *Journal of Applied Ecology* **51**, 1450-1459, doi:10.1111/1365-2664.12306 (2014).
- 9 Turner, C. R. *et al.* Particle size distribution and optimal capture of aqueous microbial eDNA. *Methods in Ecology and Evolution* **5**, 676-684, doi:10.1111/2041-210X.12206 (2014).
- 10 Hänfling, B. *et al.* Environmental DNA metabarcoding of lake fish communities reflects long-term data from established survey methods. *Molecular Ecology* **25**, 3101-3119, doi:10.1111/mec.13660 (2016).
- 11 Valentini, A. *et al.* Next-generation monitoring of aquatic biodiversity using environmental DNA metabarcoding. *Molecular Ecology* **25**, 929–942 (2015).
- 12 Ruse, L. Classification of nutrient impact on lakes using the chironomid pupal exuvial technique. *Ecological Indicators* **10**, 594-601, doi:10.1016/j.ecolind.2009.10.002 (2010).
- 13 Nguyen, N.-P., Warnow, T., Pop, M. & White, B. A perspective on 16S rRNA operational taxonomic unit clustering using sequence similarity. *Npj Biofilms And Microbiomes* **2**, 16004, doi:10.1038/npjbiofilms.2016.4 (2016).
- 14 Biggs, J. *et al.* Using eDNA to develop a national citizen science-based monitoring programme for the great crested newt (*Triturus cristatus*). *Biological Conservation* -, doi:10.1016/j.biocon.2014.1011.1029, doi:http://dx.doi.org/10.1016/j.biocon.2014.11.029 (2015).
- 15 O'Donnell, J. L., Kelly, R. P., Lowell, N. C. & Port, J. A. Indexed PCR Primers Induce Template-Specific Bias in Large-Scale DNA Sequencing Studies. *PLoS ONE* **11**, e0148698, doi:10.1371/journal.pone.0148698 (2016).
- 16 Berry, D., Ben Mahfoudh, K., Wagner, M. & Loy, A. Barcoded Primers Used in Multiplex Amplicon Pyrosequencing Bias Amplification (vol 77, pg 7846, 2011). *Applied and Environmental Microbiology* **78**, 612-612, doi:10.1128/aem.07448-11 (2012).

REVIEWERS' COMMENTS:

Reviewer #1 (Remarks to the Author):

The paper is improved mainly with regard to the more realistic estimates of species richness in these samples, and by making a comparison of DNA data with the direct identification of exuviae. I think it is a neat methodological study showing that exuviae can be obtained from the lake surface and used for DNA extractions in bulk for a rather complete species list at a point in time/space. It is also interesting to see that the same information can be obtained from the water column, although the tests of this are not very explicit. The study works out the details of the barcoding methodology, although the primers and general procedures used are probably not the best compared what is currently in the literature (one fragment too long, the other too short and too lineage specific). I remain less impressed by the wider conclusions about change through time. The main conclusion at the end "... the findings address key outstanding questions related to the ecological relevance and temporal persistence of freshwater eDNA in a natural ecosystem,..." are going much beyond what has been shown here. At best, they show that the abundances of the OTUs changes over the year (Table 1), and there are mild changes in the species richness through the year and that OTU richness is lower in the winter (Fig. 3, but probably over-fitting the model). That's hardly answering a 'key outstanding question' or anything we didn't know based on traditional studies. Certainly this doesn't address the issue of persistence of the DNA in the water column and its dependency on the temperature, which would have been a very interesting topic to address, perhaps based on discrepancies in species detection using the exuviae and water column, as long persistence may account for the rather low dip in species diversity in the winter. Other topics are rather contrived, such as the correlation of read numbers in the lake and UK-wide abundance (Fig. 4). The correlation shows huge scatter, and even if significant, it probably doesn't explain much of the variation (and wouldn't be expected to).

I also found the paper very difficult to read, partly because the sentences are not well structured, the logical connection is unclear, slightly incorrect verbs or adjectives are used, and the text is not adapted to a structure that presents the Methods at the end. For example, "Removal of OTUs present at less than 0.01% yielded equitable levels of OTU genus richness for the community DNA (37 genera) and eDNA (43 genera) according to 2014 Chironomidae records of Llyn Padarn (31 genera) (Fig. 1), and was within the limits of a small number of non-target reads detected in the positive control samples." This sentence is hardly intelligible without looking at the figure and reading the Methods section. There are many other examples. The Discussion is rather wordy and addresses general topics of metabarcoding, most of which is intuitive or already said in the literature, perhaps with the exception that the data are based on different time points rather than different localities in most existing studies, which is a rather trivial difference if it is not possible to take into account persistence time. I think the paper should be focused on the real strengths, which is the methodology of metabarcoding from exuviae (which is called 'community barcoding' for unclear reason) and the comparison with the direct sequencing from the water column, showing all relevant validation of the methodology.

Reviewer #2 (Remarks to the Author):

Dear authors, dear editor,

I have read the greatly improved version of the MS. I see that all my suggestions with respect to data analysis have been included (or if not, I agree with the rebuttal provided). I cannot comment on the GAM analyses made as I am not too familiar with GAMs and the software used.

In conclusion and as stated in the earlier review: The data are novel, highly relevant and of interest to a broad readership. The MS itself excellently written. In view of the great job done by the author team in order to revise the MS I can now fully recommend publication of this study in Nature Communications.

Kind regards
Florian Leese

Reviewer #3 (Remarks to the Author):

Re-submission of:

Bista et al. "Annual time-series analysis of aqueous eDNA reveals ecologically relevant dynamics of lake ecosystem biodiversity".

The authors have made an honest and thorough attempt to address the concerns and comments made by the reviewers – including those of my own (Rev#3). And I am happy to hear that my input has been helpful for improving the manuscript. The manuscript has severely increased in quality by re-analysing the data using more conservative (and empirical) trimming cut-off values, which has now yielded lower and (presumably) more realistic diversity proxy from eDNA. Also, the inclusion of historical data on Chironomidae diversity is a fine addition as it serves as direct comparison and quality check of the eDNA data.

Although the manuscript has significantly gained from this round of review, and the re-analyses to address the reviewer points is as plentiful as one can expect from a revision, I regret that I still think this paper would be more suited for an ecological journal such as *Ecology Letters* or *Molecular Ecology*. It is a good contribution to the eDNA field, but given the incredible amount of papers coming out on this topic (although many only of technical interest), the novelty of this study is not extremely impressive nor highly relevant to a broader scientific community.

Below is a bit more elaboration on the point of tagged primers (according to answer #46), that I hope the authors find useful.

First of all; I do not generally doubt that the sequences from this study comes mainly from the respective samples, and the fact that all negative controls remained negatives strongly indicates that cross-contamination was not an issue. However, the chance of cross-contamination inevitably increases when working with untagged PCR products in copies of

millions/billions instead of low-concentration template eDNA that is tagged from the beginning. Tag-jumps and biased amplification is surely an issue, when choosing the approach of initially tagged primers. But as far as I can see, the initial primers used here has a long Illumina tail along with a series of 5 N's (Table ST6). So there is already a bias here from the N's, which will be different across samples, right?? Also, the long Illumina tail probably also affect amplification efficiency - although in similar manner across samples. As there is probably no flawless approach in metabarcoding at this moment, it could be good with a few more sentences in the discussion on these approaches along with pros and cons.

Finally (and apologies for not noticing this previously): It might be useful to show Venn-diagrams as in Figure 1 divided into each seasons/samplings, to see how well the overlap is at each specific sampling point. This could be a figure in Supplementary Material.

REVIEWERS' COMMENTS:

Reviewer #1 (Remarks to the Author):

The paper is improved mainly with regard to the more realistic estimates of species richness in these samples, and by making a comparison of DNA data with the direct identification of exuviae. I think it is a neat methodological study showing that exuviae can be obtained from the lake surface and used for DNA extractions in bulk for a rather complete species list at a point in time/space. It is also interesting to see that the same information can be obtained from the water column, although the tests of this are not very explicit. The study works out the details of the barcoding methodology, although the primers and general procedures used are probably not the best compared what is currently in the literature (one fragment too long, the other too short and too lineage specific).

1. We very much appreciate that the reviewer acknowledges how the manuscript has been improved and pinpoints aspects that they are interested in. With respect, the use of the long and short fragment was a focal part of the study and yield, in our opinion, very useful insights into the nature of eDNA in the wild. Moreover, many eDNA studies use much shorter fragments than the short fragment used here (many examples in ¹) and Supplementary Figures 3 and 4 in no way suggest that the primers that amplify the shorter fragment are too lineage specific.

I remain less impressed by the wider conclusions about change through time. The main conclusion at the end "... the findings address key outstanding questions related to the ecological relevance and temporal persistence of freshwater eDNA in a natural ecosystem,..." are going much beyond what has been shown here. At best, they show that the abundances of the OTUs changes over the year (Table 1), and there are mild changes in the species richness through the year and that OTU richness is lower in the winter (Fig. 3, but probably over-fitting the model). That's hardly answering a 'key outstanding question' or anything we didn't know based on traditional studies. Certainly this doesn't address the issue of persistence of the DNA in the water column and its dependency on the temperature, which would have been a very interesting topic to address, perhaps based on discrepancies in species detection using the exuviae and water column, as long persistence may account for the rather low dip in species diversity in the winter. Other topics are rather contrived, such as the correlation of read numbers in the lake and UK-wide abundance (Fig. 4). The correlation shows huge scatter, and even if significant, it probably doesn't explain much of the variation (and wouldn't be expected to).

2. Again, with respect, we do believe that our manuscript addresses key outstanding questions in relation to the "ecology of DNA"². Namely, focusing on the persistence of eDNA in a natural ecosystem and to our knowledge, this has never been demonstrated before. We agree that investigations into the effect of temperature will be interesting, but this will likely require experimental manipulation in model systems in order to robustly test hypotheses. The correlation of read numbers and UK-wide abundance aims to provide a broad comparison

according to UK historical records and as the reviewer states, the significance demonstrates the association.

I also found the paper very difficult to read, partly because the sentences are not well structured, the logical connection is unclear, slightly incorrect verbs or adjectives are used, and the text is not adapted to a structure that presents the Methods at the end.

3. We are sorry that the reviewer found the manuscript difficult to read. In our final revisions, we have performed text edits, in order to improve further and will work carefully with the Editorial team to implement enhancements where required. Additionally, we would like to quote reviewer 2 who suggests that the manuscripts is "excellently written".

For example, "Removal of OTUs present at less than 0.01% yielded equitable levels of OTU genus richness for the community DNA (37 genera) and eDNA (43 genera) according to 2014 Chironomidae records of Llyn Padarn (31 genera) (Fig. 1), and was within the limits of a small number of non-target reads detected in the positive control samples." This sentence is hardly intelligible without looking at the figure and reading the Methods section.

4. We have edited this sentence (Page 8, lines 189-192) and now believe it is clear when taking into account the overall context.

There are many other examples.

5. We hope that in our final revisions these have been addressed, but without the provision of examples, it is difficult to address further.

The Discussion is rather wordy and addresses general topics of metabarcoding, most of which is intuitive or already said in the literature, perhaps with the exception that the data are based on different time points rather than different localities in most existing studies, which is a rather trivial difference if it is not possible to take into account persistence time. I think the paper should be focused on the real strengths, which is the methodology of metabarcoding from exuviae (which is called 'community barcoding' for unclear reason) and the comparison with the direct sequencing from the water column, showing all relevant validation of the methodology.

6. This final comment has left us wondering, since we believe that throughout the manuscript we focus on the strengths, i.e. metabarcoding from exuviae and the comparison with the direct sequencing from the water column? For further clarity, we need to highlight that we do not refer to community barcoding, but the analysis of 'community DNA' and now provide a reference³ at first mention (Page 4, line 87) that describes the different terminologies and approaches emerging from the field.

Reviewer #2 (Remarks to the Author):

Dear authors, dear editor,

I have read the greatly improved version of the MS. I see that all my suggestions with respect to data analysis have been included (or if not, I agree with the rebuttal provided). I cannot comment on the GAM analyses made as I am not too familiar with GAMs and the software used.

In conclusion and as stated in the earlier review: The data are novel, highly relevant and of interest to a broad readership. The MS itself excellently written. In view of the great job done by the author team in order to revise the MS I can now fully recommend publication of this study in Nature Communications.

Kind regards

Florian Leese

7. We would like to express our thanks to Professor Leese and delighted that he shares our enthusiasm in the research findings.

Reviewer #3 (Remarks to the Author):

Re-submission of:

Bista et al. "Annual time-series analysis of aqueous eDNA reveals ecologically relevant dynamics of lake ecosystem biodiversity".

The authors have made an honest and thorough attempt to address the concerns and comments made by the reviewers – including those of my own (Rev#3). And I am happy to hear that my input has been helpful for improving the manuscript.

8. We are also very grateful to all the reviewers for their inputs that have greatly enhanced the quality and depth of the manuscript.

The manuscript has severely increased in quality by re-analysing the data using more conservative (and empirical) trimming cut-off values, which has now yielded lower and (presumably) more realistic diversity proxy from eDNA. Also, the inclusion of historical data on Chironomidae diversity is a fine addition as it serves as direct comparison and quality check of the eDNA data.

Although the manuscript has significantly gained from this round of review, and the re-analyses to address the reviewer points is as plentiful as one can expect from a revision, I regret that I still think this paper would be more suited for an ecological journal such as fx Ecology Letters or Molecular Ecology. It is a good contribution to the eDNA field, but given the incredible amount of papers coming out on this topic (although many only of technical interest), the novelty of this study is not extremely impressive nor highly relevant to a broader scientific community.

9. With respect, we are certain that this novel paper provides valuable insights to the field and is relevant to the broader scientific community. Moreover, our view is shared by the eminently qualified reviewer 2 and is supported by the Nature Communications Editorial Board.

Below is a bit more elaboration on the point of tagged primers (according to answer #46), that I hope the authors find useful.

First of all; I do not generally doubt that the sequences from this study comes mainly from the respective samples, and the fact that all negative controls remained negatives strongly indicates that cross-contamination was not an issue.

10. Excellent. We are delighted that Reviewer 3 acknowledges the hard work of Dr. Bista in avoiding cross contamination.

However, the chance of cross-contamination inevitably increases when working with untagged PCR products in copies of millions/billions instead of low-concentration template eDNA that is tagged from the beginning. Tag-jumps and biased amplification is surely an issue, when choosing the approach of initially tagged primers.

11. Reviewer 3 is now stating that "Tag Jumping" (one of their initial concerns?) and biased amplification is an issue with initially tagged primers, which we did not use?

But as far as I can see, the initial primers used here has a long Illumina tail along with a series of 5 N's (Table ST6). So there is already a bias here from the N's, which will be different across samples, right??

12. No. The reviewer here is misunderstanding the molecular biology underpinning the Illumina recommended protocol⁴ for tagging amplicons. The same degenerate primer is used across all samples. The combination of Ns is designed to optimise cluster calling in the early cycles of Illumina sequencing, but the Ns are the same across all samples.

Also, the long Illumina tail probably also affect amplification efficiency - although in similar manner across samples.

13. The reviewer may be correct but the point here is that the tail is the same across samples. The reviewer's preferred approach is to use different primers to amplify different samples which has been empirically proven to skew beta diversity patterns^{5,6}; this is something we wanted to avoid.

As there is probably no flawless approach in metabarcoding at this moment, it could be good with a few more sentences in the discussion on these approaches along with pros and cons.

14. Following the first round of reviews, we have explained and justified clearly our chosen Illumina recommended strategy in the primary manuscript between lines 637-657. We believe that further dialogue on this technical issue would be misplaced in the Discussion, but would be happy to take an Editorial steer on this issue if further dialogue is required.

Finally (and apologies for not noticing this previously): It might be useful to show Venn-diagrams as in Figure 1 divided into each seasons/samplings, to see how well the overlap is at each specific sampling point. This could be a figure in Supplementary Material.

15. A good idea thank you. We have provided this as Supplementary Figure 8, and refer to it in the main text, Page 11, lines 291-292.

References:

- 1 Rees, H. C., Maddison, B. C., Middleditch, D. J., Patmore, J. R. M. & Gough, K. C. The detection of aquatic animal species using environmental DNA - a review of eDNA as a survey tool in ecology. *Journal of Applied Ecology* **51**, 1450-1459, doi:10.1111/1365-2664.12306 (2014).
- 2 Barnes, M. A. & Turner, C. R. The ecology of environmental DNA and implications for conservation genetics. *Conservation Genetics* **17**, 1-17, doi:10.1007/s10592-015-0775-4 (2016).
- 3 Creer, S. *et al.* The ecologist's field guide to sequence-based identification of biodiversity. *Methods in Ecology and Evolution*, doi:10.1111/2041-210X.12574 (2016).
- 4 Miya, M. *et al.* MiFish, a set of universal PCR primers for metabarcoding environmental DNA from fishes: detection of more than 230 subtropical marine species. *Royal Society Open Science* **2** (2015).
- 5 O'Donnell, J. L., Kelly, R. P., Lowell, N. C. & Port, J. A. Indexed PCR Primers Induce Template-Specific Bias in Large-Scale DNA Sequencing Studies. *PLoS ONE* **11**, e0148698, doi:10.1371/journal.pone.0148698 (2016).
- 6 Berry, D., Ben Mahfoudh, K., Wagner, M. & Loy, A. Barcoded Primers Used in Multiplex Amplicon Pyrosequencing Bias Amplification (vol 77, pg 7846, 2011). *Applied and Environmental Microbiology* **78**, 612-612, doi:10.1128/aem.07448-11 (2012).